# Aβ42 oligomers trigger synaptic loss through CAMKK2-AMPK-dependent effectors coordinating mitochondrial fission and mitophagy

Annie Lee[1,2,3,12], Chandana Kondapalli[1,2,12], Daniel M. Virga [1,2,4,12], Tommy L. Lewis Jr[1,2,5], So Yeon Koo[6], Archana Ashok[6], Georges Mairet-Coello[7], Sebastien Herzig[8], Marc Foretz [9], Benoit Viollet [9], Reuben Shaw [8], Andrew Sproul[6,10] & Franck Polleux [1,2,11] ✉

During the early stages of Alzheimer's disease (AD) in both mouse models and human patients, soluble forms of Amyloid-β 1–42 oligomers (Aβ42o) trigger loss of excitatory synapses (synaptotoxicity) in cortical and hippocampal pyramidal neurons (PNs) prior to the formation of insoluble amyloid plaques. In a transgenic AD mouse model, we observed a spatially restricted structural remodeling of mitochondria in the apical tufts of CA1 PNs dendrites corresponding to the dendritic domain where the earliest synaptic loss is detected in vivo. We also observed AMPK over-activation as well as increased fragmentation and loss of mitochondrial biomass in Ngn2-induced neurons derived from a new APP[Swe/Swe] knockin human ES cell line. We demonstrate that Aβ42o-dependent over-activation of the CAMKK2-AMPK kinase dyad mediates synaptic loss through coordinated phosphorylation of MFF-dependent mitochondrial fission and ULK2-dependent mitophagy. Our results uncover a unifying stress-response pathway causally linking Aβ42o-dependent structural remodeling of dendritic mitochondria to synaptic loss.

Alzheimer's disease (AD) is the most prevalent form of dementia in humans, leading to socially devastating cognitive defects with poorly effective treatments. AD is characterized by two hallmark lesions, including extracellular amyloid plaques composed of fibrillar forms of Amyloid-β (Aβ) and intracellular neurofibrillary tangles (NFTs) composed of aggregated, hyperphosphorylated Tau. Before the formation of amyloid plaques and NFTs, amyloidogenic processing of amyloid precursor protein (APP) by β- and γ-secretase produces abnormal accumulation of a 42-amino acid-long amyloid-beta (Aβ42) peptide. Aβ42 peptides have a strong ability to oligomerize and form dimers, trimers, and higher order oligomers that ultimately fibrillate to form Aβ plaques[1]. Oligomeric forms of Aβ42 (Aβ42o) lead to early loss of

[1]Department of Neuroscience, Columbia University Medical Center New York, New York, NY, USA. [2]Mortimer B. Zuckerman Mind Brain Behavior Institute, New York, NY, USA. [3]The Integrated Graduate Program in Cellular, Molecular, and Biomedical Studies, Columbia University Medical Center, New York, NY, USA. [4]Department of Biological Sciences, Columbia University, New York, NY, USA. [5]Aging & Metabolism Program, Oklahoma Medical Research Foundation, Oklahoma City, OK, USA. [6]Taub Institute for Research on Alzheimer's Disease and the Aging Brain, Columbia University Medical Center, New York, NY, USA. [7]UCB Biopharma, Braine l'Alleud, Belgium. [8]Molecular & Cell Biology Laboratory, Salk Institute for Biological Studies, La Jolla, CA, USA. [9]Institut Cochin, Université de Paris, CNRS, INSERM, Paris, France. [10]Department of Pathology and Cell Biology, Columbia University Medical Center, New York, NY, USA. [11]Kavli Institute for Brain Sciences, Columbia University Medical Center, New York, NY, USA. [12]These authors contributed equally: Annie Lee, Chandana Kondapalli, Daniel M. Virga. ✉e-mail: fp2304@columbia.edu

excitatory synapses (synaptotoxic effects) in cortical and hippocampal pyramidal neurons (PNs) before plaque formation, strongly suggesting that synaptotoxicity is an early event in the disease progression, triggered by soluble Aβ42o[2–6]. This phenotype has been observed in various AD mouse models, including the J20 transgenic mouse model (hAPP[Swe,Ind]), where progressive loss of excitatory synaptic connections occurs prior to Aβ plaque formation[2,7,8]. Soluble Aβ42o, extracted biochemically from AD patients or produced synthetically, also leads to rapid synaptic loss in PNs in vitro[2,5,6,9–11].

A second phenotype observed at the early stages of AD progression is structural abnormalities of mitochondria. Because of the terminal postmitotic status and extreme degree of compartmentalization of neurons of the central nervous system (CNS), maintaining mitochondrial homeostasis in neurons is highly dependent on the dynamic balance of fission/fusion and the efficient autophagic degradation of aged and damaged mitochondria[12–17]. Mitochondrial fission is carried out by a GTPase protein called Drp1 that needs to be recruited to the outer mitochondrial membrane (OMM) by various proteins, including MFF, Fis1, MiD49, and MiD51. The fusion machinery includes Mfn1/2, which is involved in the fusion of the OMM, and OPA1, which is involved in inner mitochondrial membrane fusion[18–20]. PNs show strikingly compartmentalized morphology of their mitochondria in the somatodendritic versus axonal domains[14]. In most cells, mitochondria play critical physiological functions that might differ between dendrites and axons, such as ATP production, $Ca^{2+}$ buffering, and lipid biosynthesis[13,14,21]. Recent evidence points to significant structural remodeling of dendritic mitochondria in hippocampal neurons in multiple AD mouse models and in AD patients[18,22–27], the hippocampus being one of the earliest affected brain regions in AD in mouse models and in AD patients[28–33]. However, the role played by changes in mitochondria structure and/or function in the disease's progression in vivo, especially in relation to synaptic loss, remains untested. Moreover, the molecular mechanisms causally linking changes in mitochondria structure/function and synaptic loss remain unknown in the context of AD pathophysiology.

Results from our lab, as well as others, have demonstrated that, in hippocampal and cortical PNs, Aβ42 oligomers overactivate AMP-activated kinase (AMPK) in a $Ca^{2+}$/calmodulin kinase kinase protein 2 (CAMKK2)-dependent manner, and preventing CAMKK2 or AMPK overactivation, either pharmacologically or genetically, protects hippocampal neurons from Aβ42o-dependent synaptic loss in vitro and in vivo in the hAPP[Swe,Ind] transgenic mouse model (J20)[2,34–36]. Importantly, AMPK is overactivated in the brain of AD patients where catalytically active AMPK accumulates in pyramidal neurons of the cortex and hippocampus[37,38].

In nonneuronal cells, AMPK is a key metabolic sensor that is catalytically activated upon increasing levels of AMP/ADP (when ATP levels drop) and regulates various downstream effectors involved in maintaining mitochondrial integrity in part via two recently identified downstream effectors: mitochondrial fission factor (MFF) and Unc-51 like autophagy activating kinases 1 and 2 (ULK1/2)[39]. MFF is an outer mitochondrial protein with two identified AMPK phosphorylation sites. Upon AMPK activation, MFF enhances the recruitment of the fission protein Drp1, which is important for efficient fragmentation of mitochondria[40], a step thought to be required for efficient mitophagy through adaptation of mitochondria size to autophagosomes. The second, coordinated step regulated by AMPK activation is through the ULK proteins. ULK1 (also called Atg1) has multiple AMPK phosphorylation sites and functions in an evolutionarily conserved process called autophagy. Autophagy is involved in degrading cytoplasmic components via lysosomes, often under nutrient stress to obtain energy. Importantly, autophagy is the only known mechanism for degrading organelles, including mitochondria (mitophagy). Mounting evidence has also implicated altered autophagy in the pathogenesis of AD, as abnormal autophagic vacuoles (AV) build up in neurites of AD

patients, at least at late stages of the disease progression[41]. Therefore, we wanted to test if the CAMKK2-AMPK pathway could be the first pathway identified to coherently regulate the major AD phenotypes, and, more precisely, if Aβ42o-dependent overactivation of AMPK triggers synaptic loss through its ability to mediate mitochondrial structural remodeling.

In this study, we first reveal that hippocampal CA1 PNs display remarkably compartmentalized structural remodeling of dendritic mitochondria in the J20 AD mouse model in vivo: we observe the significant loss of mitochondrial biomass only in the apical tuft dendrites of CA1 PNs which, most significantly, corresponds to the dendritic compartment in the hippocampus where the loss of excitatory synapses can be first observed in this and other AD mouse models[42]. We next demonstrate that Aβ42o application induces sequential mitochondrial fission and loss of mitochondrial biomass through mitophagy in dendrites of cortical PNs via coordinated phosphorylation and activation of two critical AMPK effectors: MFF and ULK2. Importantly, individual manipulation of either MFF or ULK2 prevents Aβ42o-dependent dendritic spine loss, demonstrating a causal link between Aβ42o-dependent mitochondria structural remodeling and synaptic maintenance. Finally, we previously identified that Aβ42o-dependent AMPK phosphorylates Tau at serine 262, and preventing S262 phosphorylation by AMPK protects cortical PNs from Aβ42o-induced synaptic loss. Interestingly, we show that the expression of Tau-S262A is protective towards Aβ42o-induced dendritic mitochondrial remodeling and synaptic loss in cortical PNs. Our results (1) demonstrate that the CAMKK2-AMPK stress-response pathway mediates Aβ42o-dependent synaptic loss through Tau-S262 and MFF-ULK2-dependent remodeling of dendritic mitochondria, (2) provide the first "unifying" stress-response pathway triggering many of the cellular defects reported in AD mouse models and in the brain of AD patients during the early stages of the disease progression, such as Aβ42o-dependent Tau phosphorylation, defects in $Ca^{2+}$ homeostasis, increased autophagy, and defective mitochondrial integrity (reviewed in ref. 1) and (3) provide the first causal link between Aβ- and Tau-dependent dendritic mitochondrial remodeling and synaptic loss.

## Results

### Spatially restricted structural remodeling of mitochondria and synaptic loss in apical tufts dendrites of CA1 hippocampal PNs in the APP[Swe, Ind] mouse model

The J20 mouse line is a powerful model to study the effect of amyloidosis in AD pathogenesis, as it expresses the human Amyloid Precursor Protein (APP) carrying two mutations found in familial forms of AD (APP[Swe,Ind] i.e., KM670.NL and V717F called J20 model thereafter), and shows Aβ42o accumulation in the hippocampus[8]. For this study, we chose to focus on the analysis of mitochondrial morphology and dendritic spine density in CA1 hippocampal neurons, which are among the first and the most affected in AD patients and AD mouse models[33,42–44]. Using sparse in utero hippocampal or cortical electroporation (E15 to P21), we were able to visualize dendritic mitochondria morphology and spine density in optically isolated dendritic segments of hippocampal CA1 PNs or layer 2/3 pyramidal neurons in vivo (Supplementary Fig. S1). Interestingly, in WT mice, both fluorescent labeling approaches[14] and serial EM reconstructions[45] demonstrate that layer 2/3 cortical PNs display elongated and fused dendritic mitochondria, forming a complex network throughout the entire dendritic arbor (Supplementary Fig. S1a, f, h). In sharp contrast, CA1 PNs display a high degree of compartmentalization of mitochondrial morphology in dendrites of WT mice: in basal and apical oblique dendrites, mitochondria are mostly small and punctate, whereas the distal apical dendrites contain elongated and fused mitochondria (Supplementary Fig. S1a–c, e, g). The transition between these two types of mitochondrial morphologies is sharp and corresponds to the boundary between the hippocampal layer stratum radiatum (SR; receiving

primarily inputs from CA3 PNs; Supplementary Fig. S1b) and stratum lacunosum moleculare (SLM; receiving presynaptic inputs from the medial entorhinal cortex MEC directly; Supplementary Fig. S1b).

We applied the same experimental approach in both WT and J20 littermate mice, and examined mitochondrial morphology and spine density in 3-month-old mice, a time point where abundant Aβ42o can be detected, but preceding appearance of amyloid plaques[8]. This analysis revealed that, consistent with what we observed in the hippocampus of WT mice at P21 (Supplementary Fig. S1), 3-month-old WT and J20 mice both display mostly small, punctate mitochondria, visualized by labeling the mitochondrial matrix, in basal and apical oblique dendrites (Fig. 1a). This is accompanied by a trending, though not significant, decrease in spine density in basal and apical oblique dendrites in the J20 mice (Fig. 1a, c). However, the distal apical dendrites of the J20 mice showed a dramatic and significant reduction of both mitochondrial length and density/biomass, as well as a significant decrease in spine density compared to WT littermates (Fig. 1a, d, e). These results reveal (1) a striking and unique degree of compartmentalization of mitochondrial morphology in dendrites of CA1 PNs not observed in cortical layer 2/3 PNs, and (2) a spatially restricted loss of mitochondrial biomass and spine density specifically in the apical tuft dendrites of CA1 PNs in the J20 mouse model. This co-occurrence of mitochondrial remodeling and decreased spine density raised the possibility, tested below, of a potential causal relationship between Aβ42o-dependent mitochondrial remodeling and synaptic maintenance.

### Aβ42o induces synaptic loss and dendritic mitochondrial remodeling in the same time frame

In order to examine the temporal relationship between mitochondrial remodeling and decreased spine density, we adopted an in vitro model allowing us to control the timing of Aβ42o application. We turned to long-term in vitro cortical PNs cultures[2], which are synaptically connected and display uniformly fused mitochondria throughout their dendritic arbor in vitro (Fig. 2) as observed in vivo (Supplementary Fig. S1). We observed significant, time-dependent changes in dendritic mitochondrial structure following acute Aβ42o application, including both a reduction of their length and a reduction in mitochondrial density/biomass (lower panels in Fig. 2a–c and Fig. 2d, e). In the same neurons, we observed a progressive reduction in dendritic spine density in Aβ42o-treated neurons between 14 and 24 h following Aβ42o application (upper panels in Fig. 2a–c and Fig. 2f). No significant changes in mitochondria morphology were observed in the soma or the axon of the same neurons or, as previously reported, when control inverted peptide (INV42) were applied at similar concentrations as Aβ42o (300–450 nM; Supplementary Fig. S2)[2,5,6,10,11]. We previously reported that, at this dose and duration, Aβ42o application does not affect neuronal viability, strongly suggesting that the effects we observe are not a secondary consequence of compromising neuronal survival[2]. Taken together, our results highlight that Aβ42o application triggers a significant degree of time-dependent, structural remodeling of dendritic mitochondria, including a reduction of mitochondrial biomass concomitant with a reduction in spine density, which is a reliable index of excitatory synapse number in dendrites of PNs[46,47].

### Aβ42o induces mitochondrial structural remodeling and AMPK overactivation in human pyramidal neurons

These results show that in vivo exposure of CA1 hippocampal PNs to Aβ42o resulting from overexpression of APP[Swe,Ind] in the J20 AD mouse model, or direct application of synthetic Aβ42o at 300–450 nM for 24 h in mature cortical PNs in vitro, leads to striking loss of dendritic mitochondrial biomass and to synaptotoxicity. However, we wanted to overcome two potential limitations of these approaches: (1) both use mouse models, which could differ from the way human neurons might respond to Aβ42o, and (2) both approaches rely on exposure of cortical or hippocampal PNs to high levels of Aβ42o. In order to address both points, we generated a novel human embryonic stem cell (hESC) APP[Swe/Swe] homozygous knockin model. CRISPR-Cas9-mediated genome engineering was used to introduce the APP[Swe] mutation into both alleles of the H9 hESC cell line (WA09, WiCell). Previous knockin of the Swedish mutation into a human PSC line demonstrated increased endogenous APP processing and generation of Aβ42o in an allelic dose-dependent manner[48].

APP[Swe/Swe] knockin hESCs and their isogenic H9 parent control line were transdifferentiated into glutamatergic cortical neurons (iNs) using doxycycline-inducible Neurogenin2 (Ngn2) expression[49]. Following efficient Ngn2-P2A-EGFP-induced neuronal differentiation (see "Methods"), we transiently transfected these neurons in a sparse manner with a mitochondrial fluorescent reporter (mito-DsRed) and quantified dendritic mitochondrial morphology in control H9 and isogenic APP[Swe/Swe]-derived neurons (Supplementary Fig. S3). Our results demonstrate a significant reduction in dendritic mitochondrial density (Supplementary Fig. S3b) and mitochondrial length (Supplementary Fig. S3c) in APP[Swe/Swe] human neurons compared to isogenic control neurons. As previously observed in cortical and hippocampal mouse neurons in vitro and in vivo[2], we also found increased AMPK activation in APP[Swe/Swe] neurons compared to isogenic control neurons, as measured biochemically by the ratio of phosphorylated AMPK-T172 to total AMPK (Supplementary Fig. S3d, e). These results demonstrate that endogenous levels of Amyloid-β generated by the processing of mutated APP[Swe] in human neurons can trigger dendritic mitochondrial remodeling and loss of biomass as well as AMPK overactivation as observed in mouse cortical and hippocampal neurons in vitro and in vivo, respectively.

### Acute Aβ42o application induces local mitophagy in dendrites

We next tested whether the loss of mitochondrial biomass induced by Aβ42o is triggered in dendrites of PNs by local mitophagy. To that end, we examined the dynamics of autophagosomes, lysosomes, and autolysosomes, as well as changes in mitochondrial biomass in cortical PNs upon Aβ42o treatment, using sparse ex utero electroporation of plasmids encoding mito-mTagBFP2, LAMP1-mEmerald, and RFP-LC3 to visualize mitochondria, lysosomes, and autophagosomes, respectively. Cortical PNs were cultured for at least 21DIV in high-density cultures, when they display mature synapses[2], and imaged live using time-lapse confocal microscopy every 15 min for 14 h following treatment with either control (vehicle) or Aβ42o (Fig. 3).

In dendrites of control cortical PNs, we observed little to no change in mitochondrial biomass or in the dynamics of LC3+ (autophagosomes), LAMP1+ (lysosomes), or LC3+/LAMP1+ (autolysosomes) vesicles over the 14 h of time-lapse imaging (Fig. 3a–d).

In contrast, upon Aβ42o treatment, we observed a significant accumulation of both LC3+ and LAMP1+ puncta in dendrites over time (Fig. 3a–d). In dendrites exposed to Aβ42o, many of the LC3+ puncta become co-localized with LAMP1+, suggesting these autolysosomes can accumulate locally in dendrites. LC3+/LAMP1+ autolysosomes do form occasionally in control dendrites (Fig. 3a–d and quantified in Fig. 3e). As quantified in Fig. 3e, treatment with Aβ42o leads to an almost fivefold increase in the percentage of dendritic segments with increased LC3+/LAMP1+ autolysosome accumulation compared to control-treated PNs. Both LC3 and LAMP1 normalized fluorescence intensity showed time-dependent increases upon Aβ42o application, highlighting local biogenesis of autophagic vacuoles and their coalescence with lysosomes in dendritic shafts upon Aβ42o treatment, respectively (Fig. 3). We also observed a significant reduction in mitochondria fluorescence (measured via mito-mTAGBFP2, (Fig. 3b–d)) in a spatially restricted area close to LC3+/LAMP1+ puncta as indicated by yellow arrows in the representative image (Fig. 3b) and the kymograph (Fig. 3c). Not all LC3+/LAMP1+ puncta correlated with loss of mitochondria within 14 h of time-lapse imaging, however, as indicated by the red arrow in the representative image and kymograph

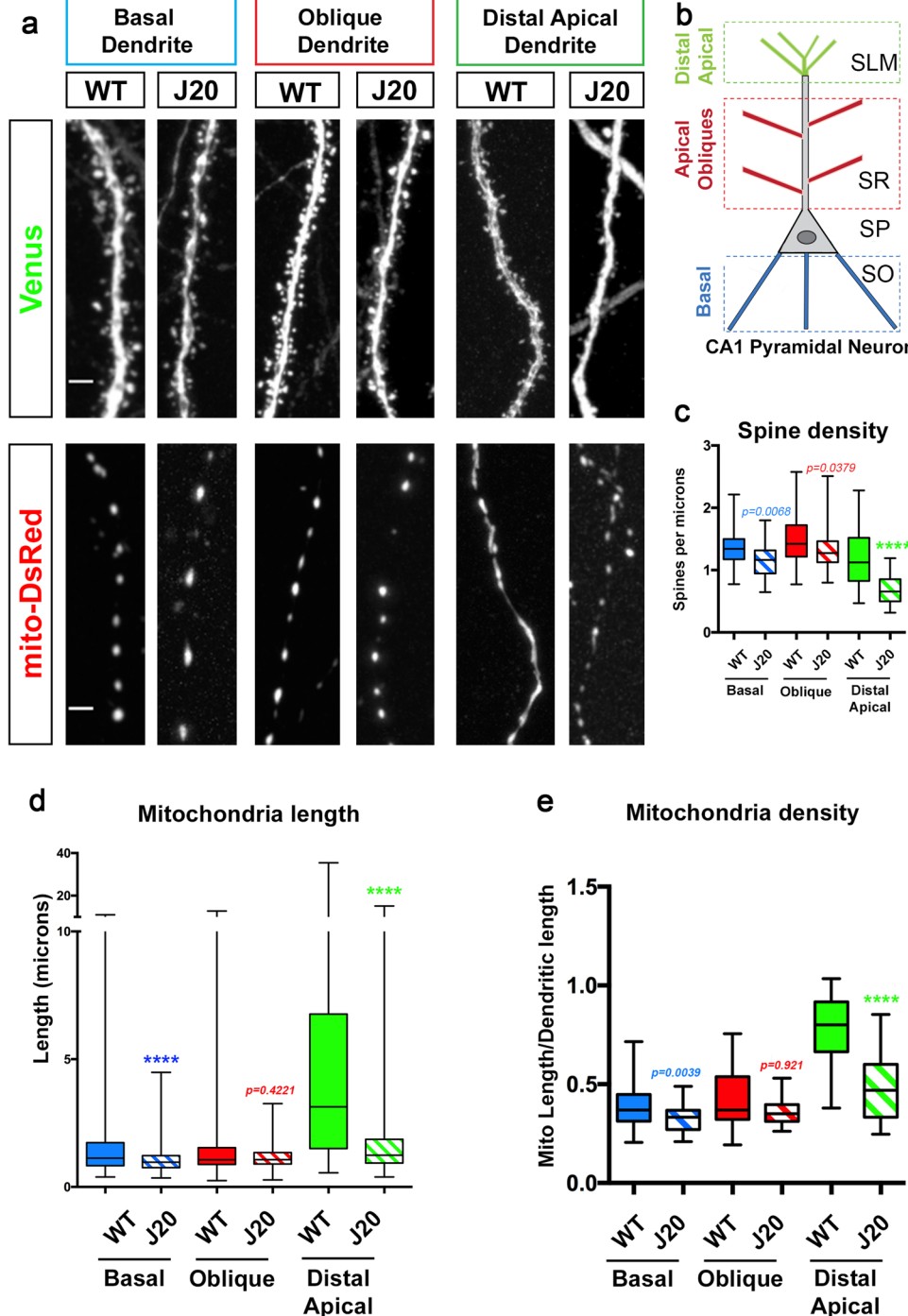

**Fig. 1 | In an AD mouse model (J20), CA1 pyramidal neurons display spatially restricted loss of mitochondrial biomass and dendritic spine loss in distal apical tufts. a, b** Representative images of CA1 PN dendritic segments from different layers of WT control and J20 mice. CA1 PNs were electroporated with pCAG-Venus and pCAG-mito-DsRed by in utero electroporation in E15.5 WT and J20 mouse embryos. Both mouse groups were fixed and imaged at 3-months postnatal (P90). Dendrites from basal, oblique, and distal apical are magnified to show spine density and mitochondrial morphology. The three types of dendritic segments were imaged in three distinct layers as depicted in **b** for both WT and J20 mice. **c–e** Quantification of spine density (**c**), individual mitochondrial length (**d**), and mitochondrial density (**e**), measured as the fraction of a dendritic segment length

occupied by mitochondria. In panels **c–e**, data are represented by box plots displaying minimum to maximum values, with the box denoting 25th, 50th (median), and 75th percentiles from at least three independent in utero electroporated mice. $n_{Basal\ WT}$ = 47 dendrites, 595 mitochondria; $n_{Basal\ J20}$ = 32 dendrites, 495 mitochondria; $n_{Oblique\ WT}$ = 46 dendrites, 630 mitochondria; $n_{Oblique\ J20}$ = 33 dendrites, 507 mitochondria; $n_{Distal\ WT}$ = 39 dendrites, 284 mitochondria; $n_{Distal\ J20}$ = 32 dendrites; 426 mitochondria. All of the analyses were done blind to the experimental conditions and were manually counted using FIJI. Statistical analyses were performed using Mann–Whitney test (**c–e**). Exact *P* values are indicated on the figure when available through Prism software, otherwise, the test significance is provided using the following criteria: ***$P < 0.001$; ****$P < 0.0001$. Scale bar = 2 μm.

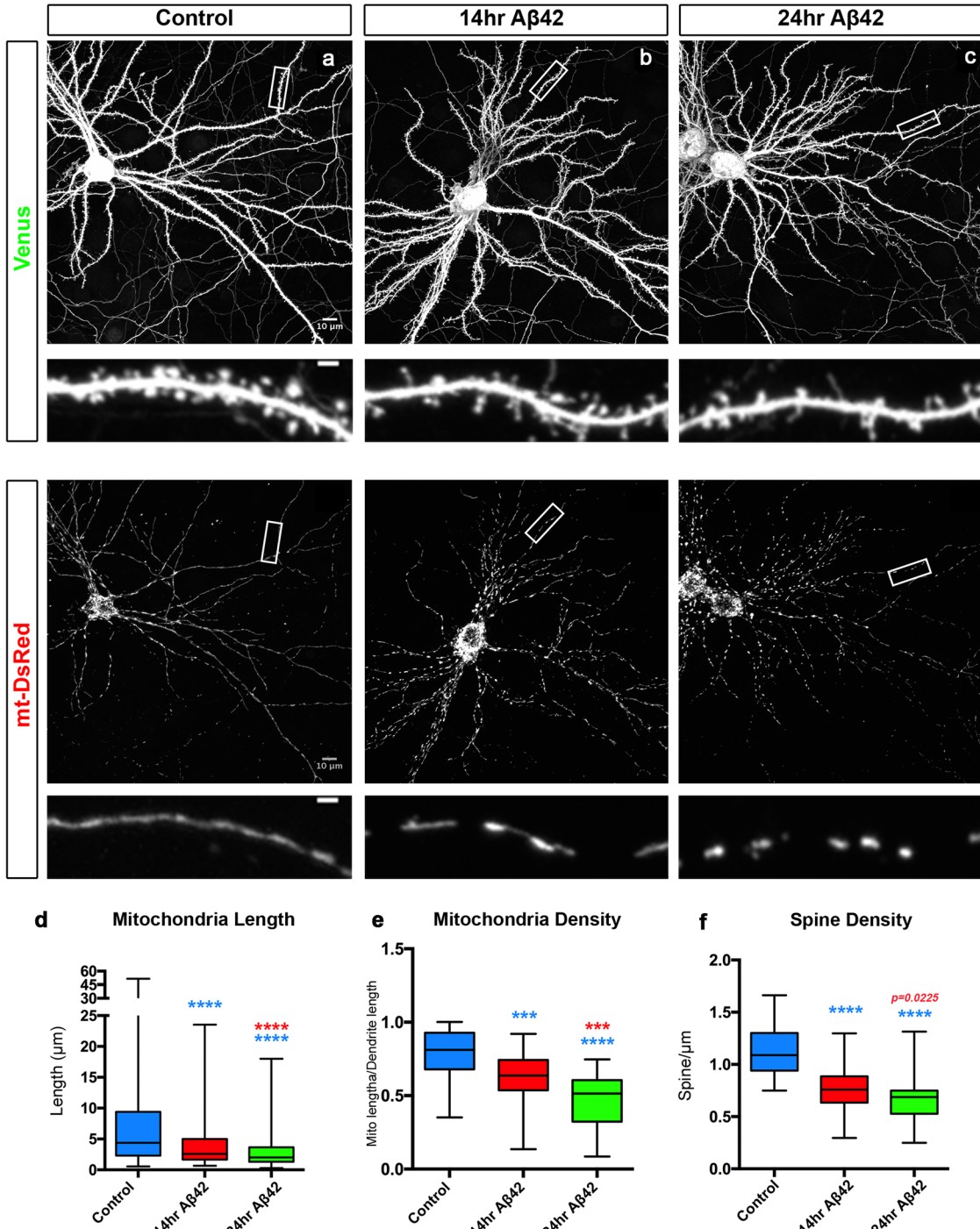

**Fig. 2 | Aβ42o induces dendritic mitochondrial fragmentation and dendritic spine loss within the same time frame. a–c** Representative images of primary cortical neurons at 21 days in vitro (DIV). Neurons express via ex utero electroporation pCAG-mVenus (upper panels in **a**–**c**) and pCAG-mito-DsRed (lower panels in **a**–**c**) to assess spine density and mitochondrial morphology, respectively. At 20 DIV, neurons were treated with either a vehicle control for 24 h or with Aβ42o (300–450 nM, see "Methods" for details) for either 14 or 24 h. High magnification of secondary dendrites is shown below the low magnification of the whole neurons. **d** Quantification of dendritic mitochondrial length. **e** Quantification of dendritic mitochondrial density. Dendritic mitochondrial density was calculated by summing the length of all the mitochondria then dividing that cumulative length by the length of the dendritic segment in which the mitochondria were quantified. **f** Quantification of dendritic spine density, calculated by dividing the number of

spines by the length of the dendrite segment in which the spines were quantified. All of the analyses were done blind to the experimental conditions and were done by manual counting using FIJI. In panels **d**–**f**, data are represented by box plots displaying minimum to maximum values, with the box denoting 25th, 50th (median), and 75th percentiles from three independent experiments. $n_{control} = 53$ dendrites, 278 mitochondria; $n_{14hr\ Aβ42o} = 47$ dendrites; 380 mitochondria; $n_{24hr\ Aβ42o} = 51$ dendrites, 400 mitochondria. Statistical analyses were performed using Mann–Whitney test in (**d**–**f**). Exact $P$ values are indicated on the figure when available through Prism software, otherwise, the test significance is provided using the following criteria: ***$P < 0.001$; ****$P < 0.0001$. Colors of significance symbols correspond to groups being compared. Scale bar for high-magnification dendritic segments = 2 μm.

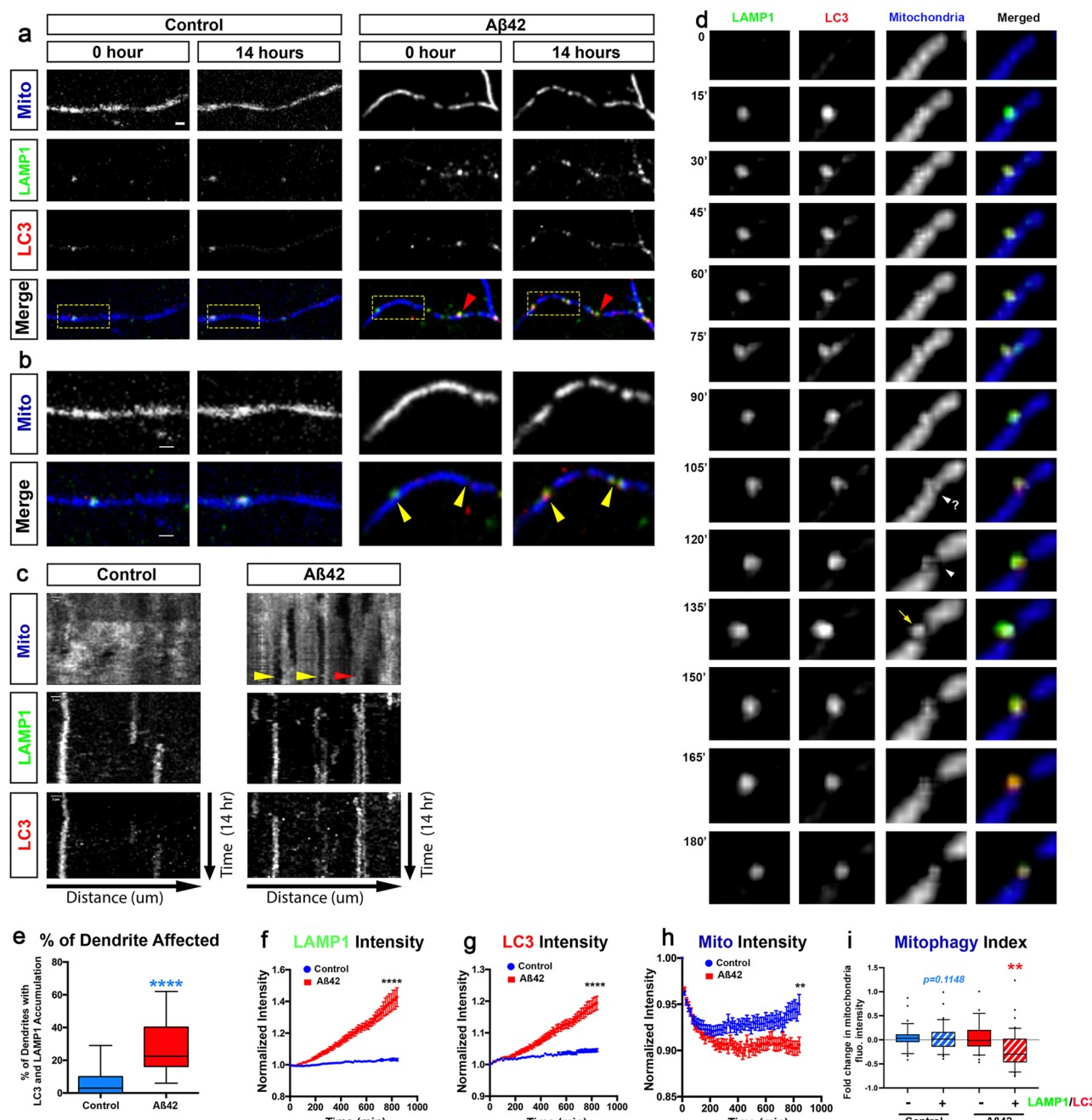

**Fig. 3 | Aβ42o treatment induces local mitophagy in dendrites. a, b** Secondary dendritic segments from primary cortical PNs at 21–25 DIV, ex utero electroporated at E15.5 with pCAG-mito-mTagBFP2, pCAG-LAMP1-mEmerald, and pCAG-RFP-LC3 to visualize mitochondria, lysosomes, and autophagosomes, respectively (**a**). At 21 DIV, neurons were treated with either a vehicle control or Aβ42o and imaged live using time-lapse microscopy every 15 min for 14 h. The yellow boxes label the magnified areas shown in **b** illustrating that, in control-treated neurons, there is no significant change in LAMP1 + (lysosomes), LC3 + (autophagosomes), or LC3-LAMP1 double-positive (autolysosomes) vesicle dynamics and/or accumulation (see Supplementary Movies S1 and S2). In contrast, in dendrites of Aβ42o-treated cortical PNs, sites of LAMP1-LC3 double-positive autolysosome accumulation often result in loss of mitochondria as indicated by yellow arrows in panel **b**. **c** Representative kymographs of mitochondria, lysosomes, and autophagosomes from dendrites in panel **a**. The yellow arrows point to loss of mitochondrial fluorescent signal, deemed mitophagy events. The red arrow indicates no decrease in mitochondrial signal, despite LAMP1 and LC3 signal, indicating an incomplete mitophagy event. **d** Detail of an autolysosome engulfing a fragment of mitochondria (135′) following a mitochondrial fission event (105′–120′). **e** The percentage of dendritic segments showing accumulation of both LC3 + autophagosomes, LAMP1 + lysosomes, or LC3-LAMP1 double-positive autolysosomes. Data are represented by box plots displaying minimum to maximum values, with the box denoting 25th, 50th (median), and 75th percentiles from three independent experiments. $n_{control} = 15$ neurons; $n_{Aβ42o} = 24$ neurons. **f–i** Quantification of **f** LAMP1 intensity, **g** LC3 intensity, and **h** Mitochondrial intensity for dendritic segments over the course of the 14 h following treatments. **i** Mitophagy index defined as the change in mitochondrial fluorescence intensity in dendritic segment with (+) or without (−) LC3-LAMP1 double-positive puncta (autophagosomes) at 14 h. In panels **f–i**, data are represented as mean +/− SEM based on $n_{Control} = 33$ dendrites; $n_{Aβ42o} = 44$ dendrites. Statistical significance was performed using a Mann–Whitney test. Exact *P* values are indicated on the figure when available through Prism software, otherwise, the test significance is provided using the following criteria: \**P* < 0.01; \*\*\*\**P* < 0.0001. Scale bar = 2 μm.

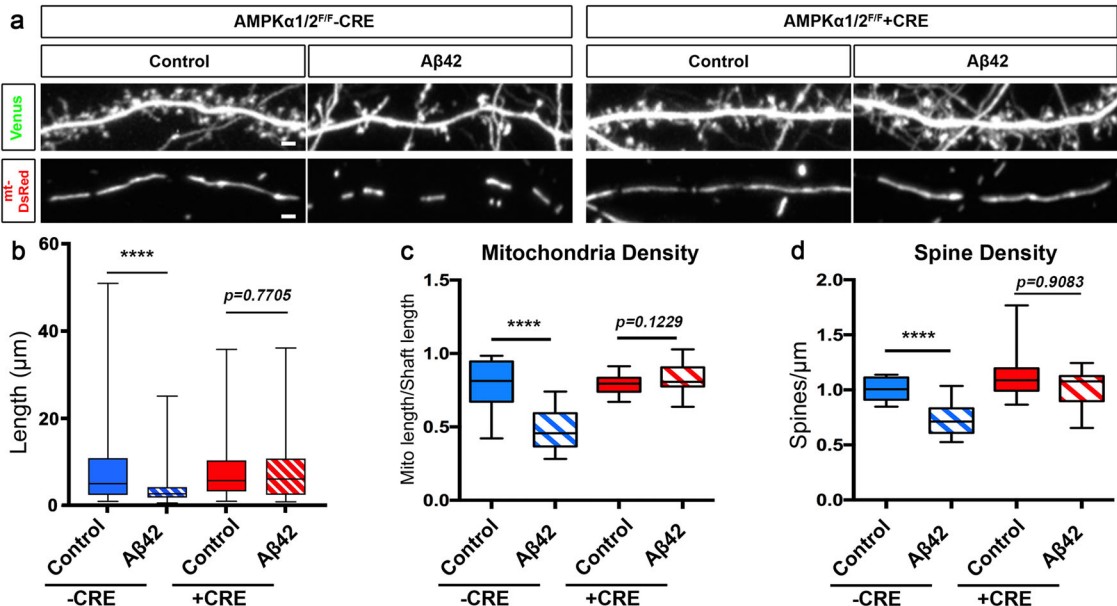

**Fig. 4 | Oligomeric Aβ42-induced synaptotoxicity and dendritic mitochondrial fragmentation is AMPK-dependent. a** Secondary dendritic segments of primary cortical PNs at 21 DIV. Embryos from AMPKα1/α2 double conditional knockout (AMPKα1$^{F/F}$/α2$^{F/F}$) were ex utero electroporated at E15.5 with pCAG-mVenus, pCAG-mito-DsRed, and either scrambled pCAG-Cre (control) or pCAG-Cre recombinase. Neurons were treated at 20 DIV with either a vehicle control or Aβ42o for 24 h. **b–d** Quantification of **b** mitochondria length, **c** mitochondrial density, and **d** spine density in individual dendritic segments. All of the analyses were done blind to the experimental conditions and were done by manual counting using FIJI. Data is represented by box plots displaying minimum to maximum values, with the box denoting 25th, 50th (median), and 75th percentiles from three independent experiments. n$_{CreNegative\ Control}$ = 27 dendrites, 128 mitochondria; n$_{CreNegative\ Aβ42o}$ = 29 dendrites, 204 mitochondria; n$_{CrePositive\ Control}$ = 29 dendrites, 143 mitochondria; n$_{CrePositive\ Aβ42o}$ = 24 dendrites, 112 mitochondria. Statistical significance was performed using a Mann–Whitney test. Exact *P* values are indicated on the figure when available through Prism software, otherwise, the test significance is provided using the following criteria: **$P < 0.01$; ****$P < 0.0001$. Scale bar = 2 μm.

(Fig. 3a–c). However, our approach of monitoring autolysosome formation and determining if they engage in mitophagy events (as opposed to autophagy of other cargoes or organelles) is validated by careful examination of these time-lapse videos. As shown in Fig. 3d, the formation and recruitment of an autolysosome (LC3+/LAMP1+) near a mitochondrion is temporally separated from fission events and actual engulfment of a mitochondrial fragment, demonstrating that these events actually correspond to mitophagosomes.

To quantify spatial dynamics of mitophagy, we defined a 'mitophagy index' along segments of dendrites by measuring the intensity of mito-mTagBFP2 fluorescence using line scan intensity measurements in regions of interest (ROI) with or without LC3+/LAMP1+ puncta. This index reflects the probability of loss of mitochondrial biomass inside versus outside of areas where autophagosomes and lysosomes coalesce to generate autolysosomes. In control conditions, the formation of LC3+/LAMP1+ puncta is not only a rare event (Fig. 3a, e), but the presence of LC3+/LAMP1+ puncta did not result in loss of mitochondria biomass, suggesting basal levels of mitophagy are low in dendrites of cortical PNs in these conditions (Fig. 3e, f). In contrast, upon Aβ42o treatment, we frequently observed spatially restricted mitophagy events (shown in Fig. 3d), only found in dendritic segments containing LC3+/LAMP1+ autolysosomes, demonstrating significant levels of mitophagy occur locally in dendrites of cortical PNs upon Aβ42o treatment (Fig. 3e–i).

## Aβ42o-dependent mitochondrial fragmentation and synaptotoxicity are mediated by CAMKK2-AMPK overactivation

We have previously shown that Aβ42o-dependent overactivation of the CAMKK2-AMPK kinase dyad mediates excitatory synaptic loss both in primary cortical PNs in culture and in hippocampal PNs in the J20 AD mouse model[2]. Since AMPK is a central regulator of mitochondrial homeostasis[39], we tested if the CAMKK2-AMPK pathway was required for

Aβ42o-induced dendritic mitochondrial remodeling in primary cortical PNs cultures in vitro using a conditional double-floxed mouse line for the two genes encoding catalytic subunits (α1 and α2) of the AMPK heterotrimers (AMPKα1$^{F/F}$/α2$^{F/F\ 50}$). We performed ex utero cortical electroporation of AMPKα1$^{F/F}$/α2$^{F/F}$ E15.5 embryos (targeting progenitors generating layer 2/3 PNs) with (1) a control plasmid lacking Cre recombinase or a plasmid expressing Cre recombinase, (2) a Venus fluorescent protein to visualize spine morphology, and (3) mito-DsRed to examine mitochondrial morphology (Fig. 4). Quantitative analysis indicated that, in basal conditions, AMPKα1/α2 null PNs have spine and mitochondrial densities indistinguishable from wild-type neurons (Fig. 4a–d). Strikingly, AMPKα1/α2 null neurons are completely protected from Aβ42o-induced mitochondrial density reduction (Fig. 4a–c) as well as dendritic spine loss (Fig. 4a–d) as previously reported[2]. This demonstrates that overactivation of AMPK is required for both Aβ42o-induced mitochondria structural remodeling and spine loss in dendrites of PNs.

AMPK has two upstream regulatory kinases, namely LKB1 and CAMKK2, which phosphorylate the AMPKα subunit on T172, significantly increasing its catalytic activity[51]. However, in cortical PNs, CAMKK2 has been identified as the primary upstream regulator of AMPK[2,35,52]. Therefore, we assessed if pharmacologically blocking CAMKK2 acutely interfered with Aβ42o effects on mitochondrial structural remodeling and dendritic spine loss. Pretreatment of primary neuronal cultures with STO-609, a specific CAMKK2 inhibitor[53], completely blocked Aβ42o-induced mitochondrial structural remodeling (Supplementary Fig. S4a–c) as well as synaptotoxicity (Supplementary Fig. S4d) as previously published[2].

## Aβ42o-induced MFF overactivation causally links mitochondrial fragmentation and synaptic loss

In nonneuronal cells, AMPK has been identified as a major metabolic sensor regulating ATP homeostasis upon induction of metabolic stress

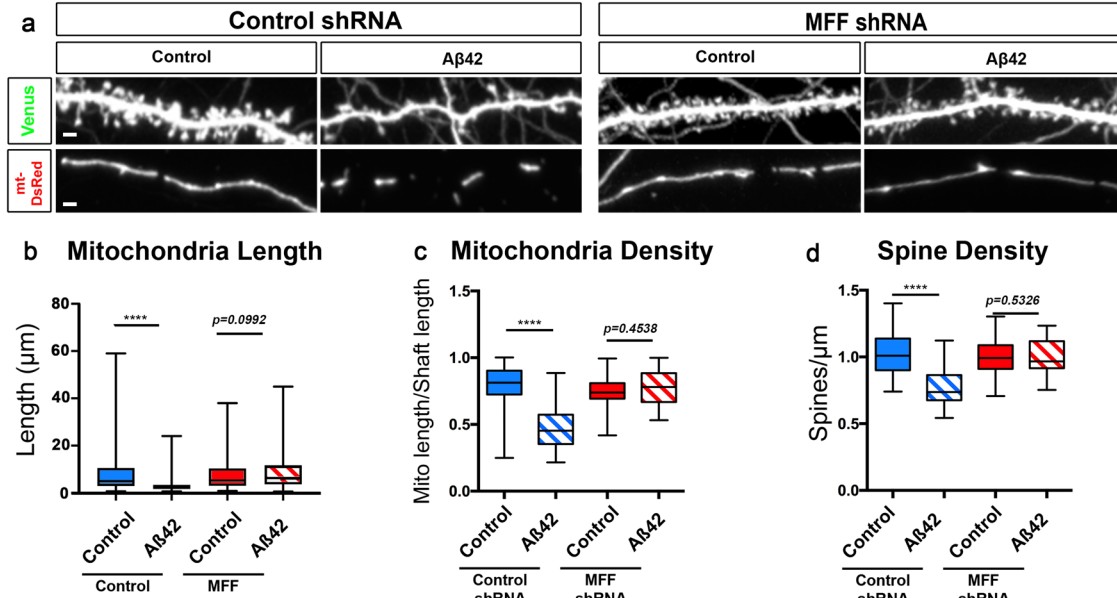

**Fig. 5 | MFF is required for Aβ42o-induced dendritic mitochondrial fragmentation and dendritic spine loss in vitro. a** Representative images of secondary dendritic segments of primary cortical PNs at 21 DIV. Embryos at E15.5 were ex utero electroporated with pCAG-mVenus, pCAG-mito-DsRed, and either with control shRNA or an shRNA specific for mouse MFF (MFF shRNA). Neurons were treated at 20 DIV with either a vehicle control or Aβ42o for 24 h. Knockdown of MFF blocks both dendritic mitochondrial fragmentation and subsequent degradation as well as dendritic spine loss. **b–d** Quantification of dendritic mitochondrial length (**b**), dendritic mitochondrial density (**c**), and dendritic spine density (**d**). All of the analyses were done blind to the experimental conditions and were done by manual counting using FIJI. Data are represented by box plots displaying minimum to maximum values, with the box denoting 25th, 50th (median), and 75th percentiles from three independent experiments. $n_{pLKO\ Control} = 36$ dendrites, 188 mitochondria; $n_{pLKO\ Aβ42o} = 41$ dendrites, 300 mitochondria; $n_{MFFshRNA\ Control} = 36$ dendrites, 167 mitochondria; $n_{MFFshRNA\ Aβ42o} = 30$ dendrites, 125 mitochondria. Statistical analyses were performed using a Mann–Whitney test in (**b–d**). Exact $P$ values are indicated on the figure when available through Prism software, otherwise, the test significance is provided using the following criteria: ****$P < 0.0001$. Scale bar for magnified dendritic segments = 2 μm.

through its ability to phosphorylate many downstream effectors involved in various aspects of metabolic homeostasis[39,52]. As part of this metabolic stress response, AMPK regulates mitochondrial fission through direct phosphorylation of mitochondrial fission factor (MFF), which enhances its ability to recruit Drp1 to the outer mitochondrial membrane and promotes mitochondrial fission[40]. To that end, we tested if MFF is the effector mediating AMPK-dependent Aβ42o-induced mitochondrial fission, and if blocking MFF-dependent fission can prevent synaptic loss. Upon exposure to Aβ42o, cortical PNs in vitro that express MFF shRNA (knockdown efficacy validated in ref. 14) are protected from loss of mitochondrial biomass in dendrites (Fig. 5a–c) and from synaptic loss (Fig. 5a, d). Strikingly, knockdown of MFF in basal conditions does not affect dendritic mitochondrial length and density (Fig. 5a–c) or dendritic spine density (Fig. 5a, d), strongly suggesting that the signaling pathway we have identified is a bona fide stress-response pathway in dendrites upon Aβ42o exposure, as we have previously hypothesized[54,55]. Altogether, these results suggest a causal relationship between Aβ42o-induced MFF-dependent dendritic mitochondrial fission and dendritic spine loss.

### MFF knockdown prevents spatially restricted mitochondrial fragmentation and reduced synaptic density in distal apical tufts in CA1 PNs of an AD mouse model in vivo

The results from the in vitro rescue experiments presented in Fig. 5 strongly suggest that blocking MFF-dependent mitochondrial fission in cortical PNs prevents spine loss mediated by acute exposure to Aβ42o. We wanted to extend this observation in vivo and for CA1 PNs. To achieve this, we performed hippocampal in utero electroporation targeting specifically CA1 PNs in wild-type or J20 (APP^(Swe, Ind)) mouse littermates[55] with plasmids expressing a mitochondrial matrix marker (mito-YFP), a cell filler (tdTomato), and either a control shRNA or an shRNA knocking down MFF[14] (Fig. 6a). As reported in Fig. 1, we observed

both a significant reduction of spine density in apical tufts dendrites of CA1 PNs and a reduction of mitochondrial volume in J20 mice compared to wild-type littermates at 3 months (Fig. 6b–d). Strikingly, shRNA-mediated knockdown of MFF prevented loss of mitochondrial volume in the apical tufts dendrites of J20 mice (Fig. 6a, c) and also rescued spine loss in the same dendritic segments (Fig. 6a, d). Because these experiments are performed at very low transfection efficiency in vivo, these rescue effects are not only cell-autonomous, they are postsynaptic-autonomous since close to 100% of the axonal inputs to these CA1 PNs where we knockdown MFF are either wild-type in the wild-type littermates or pathologically unmodified in the J20 APP-expressing mice as they have no MFF knockdown. We conclude that Aβ42o-mediated mitochondrial structural remodeling in apical tufts of CA1 PNs in vivo are causally linked to the postsynaptic loss of spines in vivo.

### Aβ42o-induced MFF phosphorylation by AMPK is required for dendritic mitochondrial fragmentation and synaptic loss

To determine if MFF is phosphorylated by Aβ42o-dependent AMPK overactivation in neurons, we measured and compared AMPK-mediated phosphorylation of MFF at serine 146 (S146; one of the AMPK phosphorylation site of Isoform 3 enriched in the brain;[56]) to total MFF in primary cortical neurons treated with Aβ42o or a control (Supplementary Fig. S5). As a positive control to show that the MFF S146 site is regulated by AMPK activation, we overexpressed the human isoform of MFF in Neuro2A (N2A) neuroblastoma cells and treated with Metformin, a potent activator of AMPK[40], or a vehicle control (Supplementary Fig. S5a, b). First, Aβ42o application for 24 h on long-term cortical PNs cultures showed a significant increase in AMPK phosphorylation at T172 site as previously reported[2,36] (Supplementary Fig. S5a, b). As a consequence of AMPK activation, we observe an increase in phosphorylation of MFF at S146 in cortical

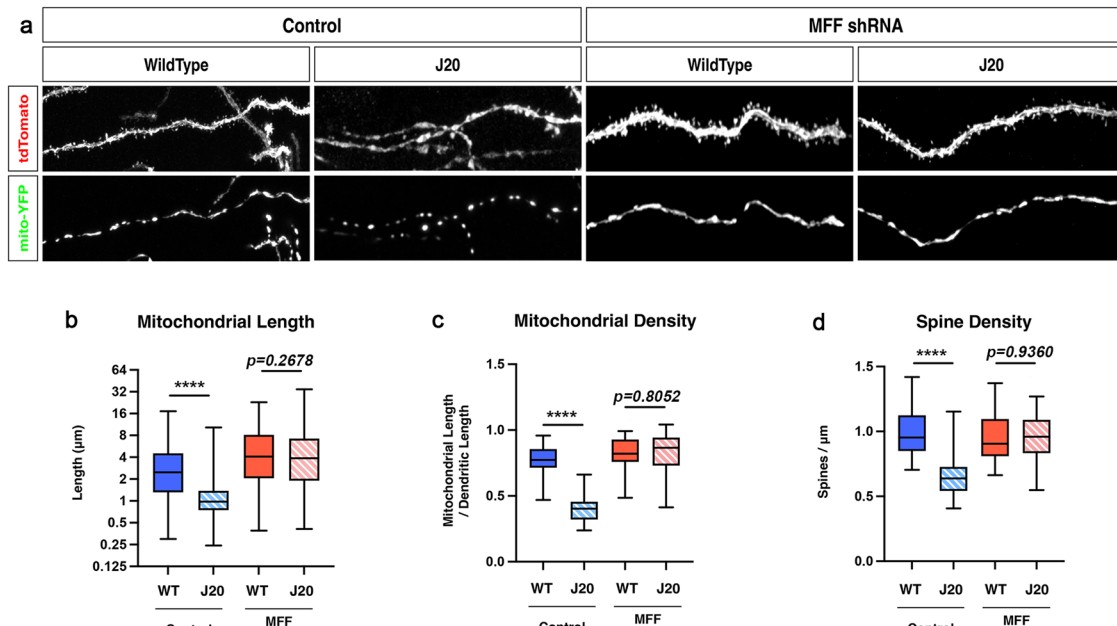

**Fig. 6 | MFF is required for Aβ42o-induced dendritic mitochondrial fragmentation and spine loss in apical tufts of CA1 PNs in vivo. a** High-magnification images of dendritic segments in stratum lacunare moleculare (SLM) of CA1 PNs of 3-months-old WT or J20 mice. CA1 PNs were co-electroporated in utero at E15.5 with plasmids expressing a cell filler (tdTomato), a mitochondrial matrix marker (mito-YFP) and either control (scrambled) shRNA or shRNA targeting mouse MFF. **b–d** Quantification of dendritic mitochondrial length (**b**), dendritic mitochondrial density (**c**), and dendritic spine density (**d**). Data are represented by box plots displaying minimum to maximum values, with the box denoting 25th, 50th (median), and 75th percentiles from at least three independent in utero electroporated mice. $n_{Control/WT}$ = 21 dendrites, 157 mitochondria; $n_{Control/J20Het}$ = 22 dendrites, 327 mitochondria; $n_{shMFF/WT}$ = 14 dendrites, 102 mitochondria; $n_{shMFF/J20Het}$ = 36 dendrites, 256 mitochondria. All of the analyses were done blind to the experimental conditions and were manually counted using FIJI. Statistical analyses were performed using a Mann–Whitney test in **b–d**. Exact *P* values are indicated on the figure when available through Prism software, otherwise, the test significance is provided using the following criteria: ****$P < 0.0001$. Scale bar for magnified dendritic segments = 2 μm.

neurons upon Aβ42o application that is comparable to Metformin treatment of N2A cells (Supplementary Fig. S5a, b). This experiment demonstrates that Aβ42o triggers phosphorylation of MFF via AMPK in cortical PNs, which is also observed in AD patients brains[37].

To test if AMPK-dependent phosphorylation of MFF is required for Aβ42o-dependent mitochondrial remodeling, we took a gene replacement approach by knocking down mouse MFF and expressing either wild-type human MFF (hMFF-WT) or human MFF that cannot be phosphorylated by AMPK (hMFF-AA) in primary cortical PNs. Low expression levels of hMFF-WT (not targeted by mouse-specific MFF shRNA) and hMFF-AA, which carries two point mutations (S155A and S172A), previously shown to abolish AMPK-mediated MFF function in mitochondrial fission[40], when combined with mouse-specific MFF shRNA, does not lead to mitochondrial fragmentation in control conditions, i.e., does not lead to MFF overexpression which was previously shown to be sufficient to induce dendritic mitochondrial fragmentation[40]. Primary cortical PNs expressing these titrated levels of hMFF-WT showed both mitochondrial and spine loss following 24 h of Aβ42o treatment (Fig. 7a–d) to levels comparable to control neurons exposed to Aβ42o, whereas neurons expressing titrated levels of hMFF-AA were protected from Aβ42o-induced mitochondrial structural remodeling and spine loss (Fig. 7a–d). These results demonstrate a causal role of Aβ42o-triggered mitochondrial remodeling, induced by AMPK-mediated phosphorylation of MFF, and synaptic loss.

## Activation of ULK2 by AMPK leads to loss of mitochondrial biomass following MFF-dependent mitochondrial fission

In nonneuronal cells under conditions of metabolic stress, AMPK activation not only triggers increased mitochondrial fission through phosphorylation and activation, but also triggers mitophagy by directly phosphorylating the autophagy initiating kinase, ULK1 (Atg1)[57].

Since we observed (1) increased mitophagy in dendrites exposed to Aβ42o, (2) a significant decrease in mitochondrial biomass following increased AMPK-MFF-dependent fission between 14–24 h of Aβ42o treatment (Figs. 2 and 3) and (3) previous results show AMPK robustly activates ULK kinases in neurons[58], we hypothesized that over-activation of AMPK by Aβ42o also mediates the observed increased levels of mitophagy by activating ULK1 and/or ULK2, two isoforms of ULK implicated in autophagy and, in particular, mitophagy and are highly expressed in developing and adult cortical neurons[57,59].

We first tested if either one or both isoforms of ULK mediate Aβ42o-induced synaptotoxicity by performing shRNA-mediated knockdown of either ULK1 or ULK2 (Supplementary Fig. S6). Interestingly, we found that knockdown of ULK2, but not ULK1, protected cortical PNs from Aβ42o-induced spine loss at 24 h of Aβ42o treatment in vitro (Supplementary Fig. S6a, c). We further validated that ULK2 knockdown can block the increased autophagy induced by Aβ42o treatment in cortical PNs: immunofluorescent detection of SQSTM1/p62 is significantly increased in the soma of cortical PNs treated with Aβ42o for 24 h compared to vehicle alone (DMSO) (Supplementary Fig. S7). Knockdown of ULK2 using shRNA blocks the increase in p62 accumulation induced by treatment with Aβ42o (Supplementary Fig. S7).

We then tested if loss of ULK2 could protect Aβ42o-treated neurons from loss of mitochondrial biomass. We sparsely expressed Cre recombinase using ex utero electroporation in primary cortical PNs isolated from a conditional ULK2 mouse line (ULK2$^{F/F}$)[60] along with Venus to visualize dendritic spines, and mito-dsRed to visualize mitochondria. Loss of ULK2 completely blocks Aβ42o-induced loss of mitochondrial biomass which is compatible with blocking ULK2-mediated mitophagy (Fig. 8a–c); interestingly, however, dendritic mitochondria remain fragmented compared to the controls (Fig. 8a–c), compatible with the Aβ42o-induced AMPK-MFF-dependent

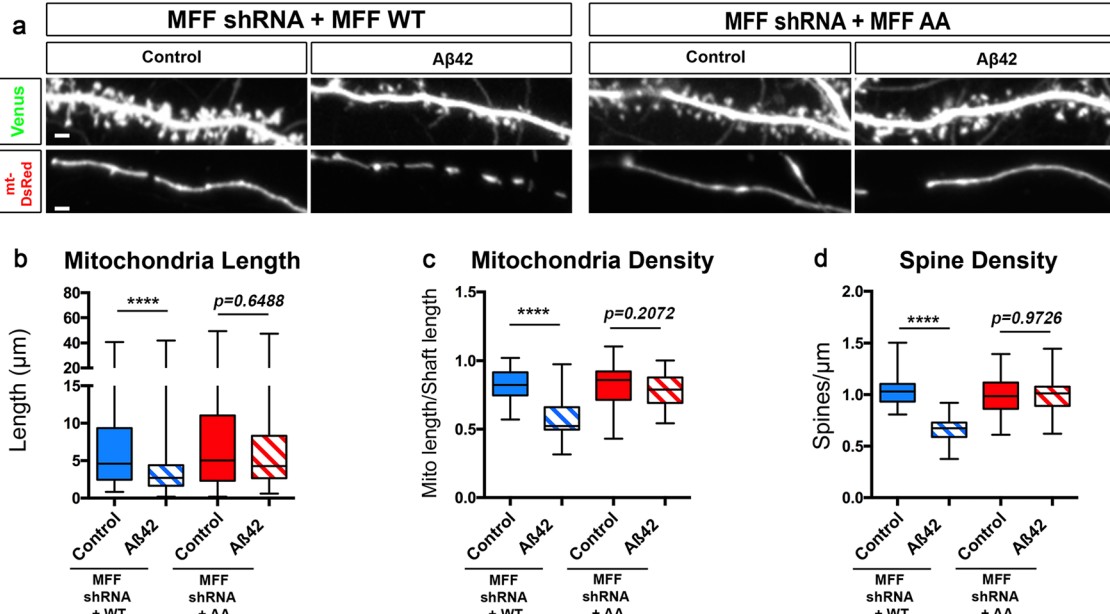

**Fig. 7 | Aβ42o induces AMPK-dependent MFF phosphorylation at two serine sites required for Aβ42o-dependent dendritic mitochondrial fragmentation and spine loss. a** Representative images of secondary dendritic segments of primary cortical PNs at 21 DIV. Embryos at E15.5 were ex utero electroporated with pCAG-Venus, pCAG-mito-DsRed, MFF shRNA, and low levels of either pCAG-MFF-WT cDNA or phospho-dead pCAG-MFF-AA. At 20DIV, the neurons were treated with either a vehicle control or Aβ42o for 24 h. Gene replacement with a form of MFF that cannot be phosphorylated by AMPK (MFF shRNA + MFF-AA) blocks dendritic mitochondrial fragmentation and subsequent spine loss. **b–d** Quantification of **b** dendritic mitochondrial length, **c** dendritic mitochondrial density, and **d** spine density. All of the analyses were

done blind to the experimental conditions and performed by manual counting using FIJI. Data are represented by box plots displaying minimum to maximum values, with the box denoting 25th, 50th (median), and 75th percentiles from three independent experiments. $n_{MFFshRNA + MFFWT\ Control}$ = 31 dendrites, 166 mitochondria; $n_{MFFshRNA + MFFWT\ Aβ42o}$ = 29 dendrites, 228 mitochondria; $n_{MFFshRNA + MFFAA\ Control}$ = 35 dendrites, 164 mitochondria; $n_{MFFshRNA + MFFAA\ Aβ42o}$ = 28 dendrites, 137 mitochondria. Statistical analyses were performed using a Mann–Whitney test in **b–d**. Exact P values are indicated on the figure when available through Prism software, otherwise, the test significance is provided using the following criteria: ****P < 0.0001. Scale bar in **a**: 2 μm.

induction of mitochondrial fission being intact in these ULK2-deficient cortical PNs. This further supports the idea that mitochondrial fission and mitophagy are coupled events, such that mitochondrial fragmentation must precede mitophagy[16,19]. Importantly, preservation of mitochondrial density/biomass upon ULK2 conditional deletion was sufficient to rescue Aβ42o-induced synaptotoxicity (Fig. 8a–d). Therefore, our data suggest that these two parallel effectors (MFF-dependent fission and ULK2-dependent mitophagy), coincidentally activated by AMPK, act in a concerted manner to couple fission and mitophagy in neuronal dendrites, and, most importantly, that loss of dendritic mitochondria biomass triggered by Aβ42o is causally linked to synaptic loss.

**AMPK phosphorylation of ULK2 is required for Aβ42o-induced dendritic mitophagy and synaptic loss**
We next tested if AMPK-mediated phosphorylation of ULK2 is required for Aβ42o-dependent synaptotoxic effects in cortical PNs. Extensive studies in nonneuronal cells have established that the closest ortholog of ULK2, ULK1, is phosphorylated and thereby catalytically activated by AMPK and plays a key role in regulating mitophagy[57,59]. Although the two orthologs seem to have redundant functions in nonneuronal cells, the above results uncovered that, in cortical PNs, only downregulation of ULK2, but not ULK1, blocks Aβ42-mediated synaptotoxicity (Supplementary Fig. S6). Therefore, we performed a sequence homology analysis of candidate AMPK phosphorylation sites identified in ULK1 and compared them to ULK2. Two independent groups collectively reported six AMPK-mediated phosphorylation sites on ULK1[57,59]. Using primary sequence alignment between human and mouse ULK1 and ULK2, we found that four of these ULK1 phosphorylation sites are conserved in ULK2 and are predicted as bona fide AMPK consensus

phosphorylation sites (Fig. 9a and Supplementary Fig. S8; S309/T441/S528/S547).

To validate that these four S/T residues are the most prevalent AMPK target sites in ULK2, we mutated these four candidate S/T residues to alanine in ULK2 (ULK2–4SA) and overexpressed them in HEK293T cells. We then performed immunoprecipitation (IP) of Flag-tagged ULK2-WT and ULK2–4SA from cells that were either treated with DMSO or AMPK direct agonist 991. Overexpression of constitutively active AMPK (ca-AMPK) in the presence of ULK2-WT or ULK2–4SA mutant conditions were used as a positive control. Immunoprecipitated Flag-ULK2-WT or Flag-ULK2–4SA were then immunoblotted with an antibody recognizing the optimal AMPK phospho-substrate motif. We observed basal levels of phosphorylation of ULK2-WT in control cells which increases upon AMPK activation via 991 treatment or when co-expressed with ca-AMPK (Fig. 9b). Importantly, AMPK-mediated phosphorylation of ULK2 was completely abolished in the 4SA mutant (Fig. 9b), strongly suggesting that these four S/T sites are the main AMPK phosphorylation sites in ULK2.

Finally, we tested if ULK2 phosphorylation by AMPK is required for the loss of mitochondrial biomass and synaptic loss induced by Aβ42o. We performed gene replacement experiments taking advantage of ULK2F/F primary cortical PNs co-electroporated with Cre and either ULK2-WT (positive control), ULK2–4SA, or kinase-inactive ULK2 (ULK2-KI, as negative control). We found that expression of ULK2-WT in ULK2-null cortical PNs enabled both loss of mitochondrial biomass and synaptic loss induced by Aβ42o, but expression of ULK2-KI or ULK2–4SA in the ULK2-null neurons resulted in no reduction in mitochondrial biomass and did prevent dendritic spine loss upon treatment with Aβ42o (Fig. 9c–f). These results confirm that AMPK-mediated phosphorylation and activation of ULK2 mediate Aβ42o-

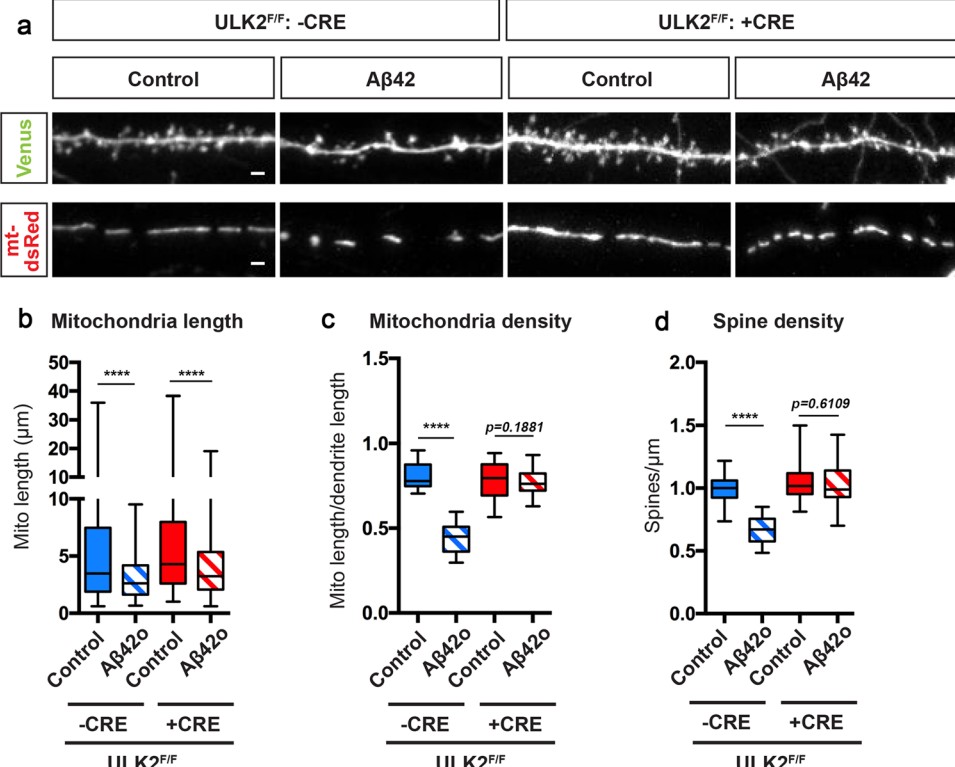

**Fig. 8 | ULK2 acts in a concerted manner with MFF and leads to loss of mitochondrial biomass following MFF-dependent mitochondrial fragmentation.**
**a** Representative images of secondary dendritic segments of primary cortical PNs at 21DIV. ULK2$^{F/F}$ embryos were ex utero electroporated at E15.5 with pCAG-Venus, pCAG-mito-DsRed, and without (-Cre) or with pCAG-Cre recombinase (+Cre). Neurons were treated at 20 DIV with either a vehicle control or Aβ42o for 24 h. Deletion of ULK2 in cortical PNs blocks Aβ42o-induced loss of dendritic spines and loss of mitochondrial biomass, but does not prevent a decrease in mitochondrial length. **b**–**d** Quantification of **b** dendritic mitochondrial length, **c** dendritic mitochondrial density, and **d** dendritic spine density. Analyses were done blind to the experimental conditions and were done by manual counting using FIJI. Data are represented by box plots displaying minimum to maximum values, with the box denoting 25th, 50th (median), and 75th percentiles from three independent experiments. $n_{CreNegative\ Control}$ = 36 dendrites, 193 mitochondria; $n_{CreNegative\ Aβ42o}$ = 38 dendrites, 254 mitochondria; $n_{CrePositive\ Control}$ = 36 dendrites, 186 mitochondria; $n_{CrePositive\ Aβ42o}$ = 35 dendrites, 435 mitochondria. Statistical analyses were performed using a Mann–Whitney test in **b**–**d**. Exact *P* values are indicated on the figure when available through Prism software, otherwise, the test significance is provided using the following criteria: ****P < 0.0001. Scale bar for magnified dendritic segments = 2 μm.

dependent synaptic loss through its ability to induce mitophagy and loss of mitochondrial biomass.

### Phosphorylation of Tau on serine 262 is required for Aβ42o-induced mitochondrial remodeling and synaptotoxicity

We and others have previously shown that AMPK phosphorylates human Tau (hTau) on the KGxS motif present at serine 262 embedded within the R1 microtubule binding domain[2,36], and that over-expressing a form of hTau that cannot be phosphorylated at S262 (hTau-S262A) blocked Aβ42o-induced synaptic loss in cortical and hippocampal PNs[2,61]. Since hTau phosphorylation has been involved in Drp1-mediated mitochondrial fission in multiple fly and mouse AD models[62,63], we tested specifically if AMPK-dependent hTauS262 phosphorylation could link Aβ42o-induced mitochondrial remodeling and synaptotoxicity. Indeed, the expression of hTau-S262A (but not wild-type hTau) protected neurons from Aβ42o-induced mitochondrial remodeling and synaptotoxicity (Supplementary Fig. S9a–d). This result strongly suggests that AMPK-dependent Tau phosphorylation on S262[2,36] participates in the MFF-ULK2-dependent mitochondrial remodeling and synaptic loss triggered by Aβ42o.

### Reduction of AMPK expression slows the rate of Amyloid-β plaque accumulation in the hippocampus of J20 mice

Finally, since (1) there is clear evidence for circuit hyperactivity at early stages in various AD mouse models (reviewed in ref. 64), (2) increased neuronal activity is sufficient to activate AMPK in a CAMKK2-dependent manner[2], and (3) amyloidogenic APP processing into Aβ42 is activity-dependent[65], we tested if part of the protective effect of reducing AMPK in the context of AD pathophysiology could be due to decreased APP processing and Aβ42 production. To do this, we monitored the accumulation of Aβ42 (using the 6E10 antibody detecting Aβ42) into plaques in wild-type, J20, AMPKα1$^{+/-}$ x J20, or AMPKα1$^{-/-}$ x J20 mice, the latter two of which have either heterozygous (AMPKα1$^{+/-}$) or homozygous (AMPKα1$^{-/-}$), constitutive knockdown of the α1 subunit of AMPK (Supplementary Fig. S11). Our results show that only complete knockout of AMPKα1 significantly reduces Aβ plaque coverage and plaque size compared to 5 months J20 mice (Supplementary Fig. S11a, b). These results suggest that AMPK overactivation observed in the J20 AD mouse model[2] participates in increased APP processing into Aβ42.

## Discussion

Loss of excitatory glutamatergic synapses in cortical and hippocampal PNs has been reported as an early event in AD progression, primarily driven by soluble Aβ42o and Tau hyperphosphorylation[5,6,8,11,66–69]; reviewed in ref. 1). This loss of connectivity is thought to be responsible for the early cognitive defects characterizing AD patients, including defective learning and memory[1,70]. Recently, mitochondrial remodeling and dysfunction in PNs have emerged as prominent cellular phenotypes in AD[24,25,27,37,71–73]. However, whether structural remodeling of

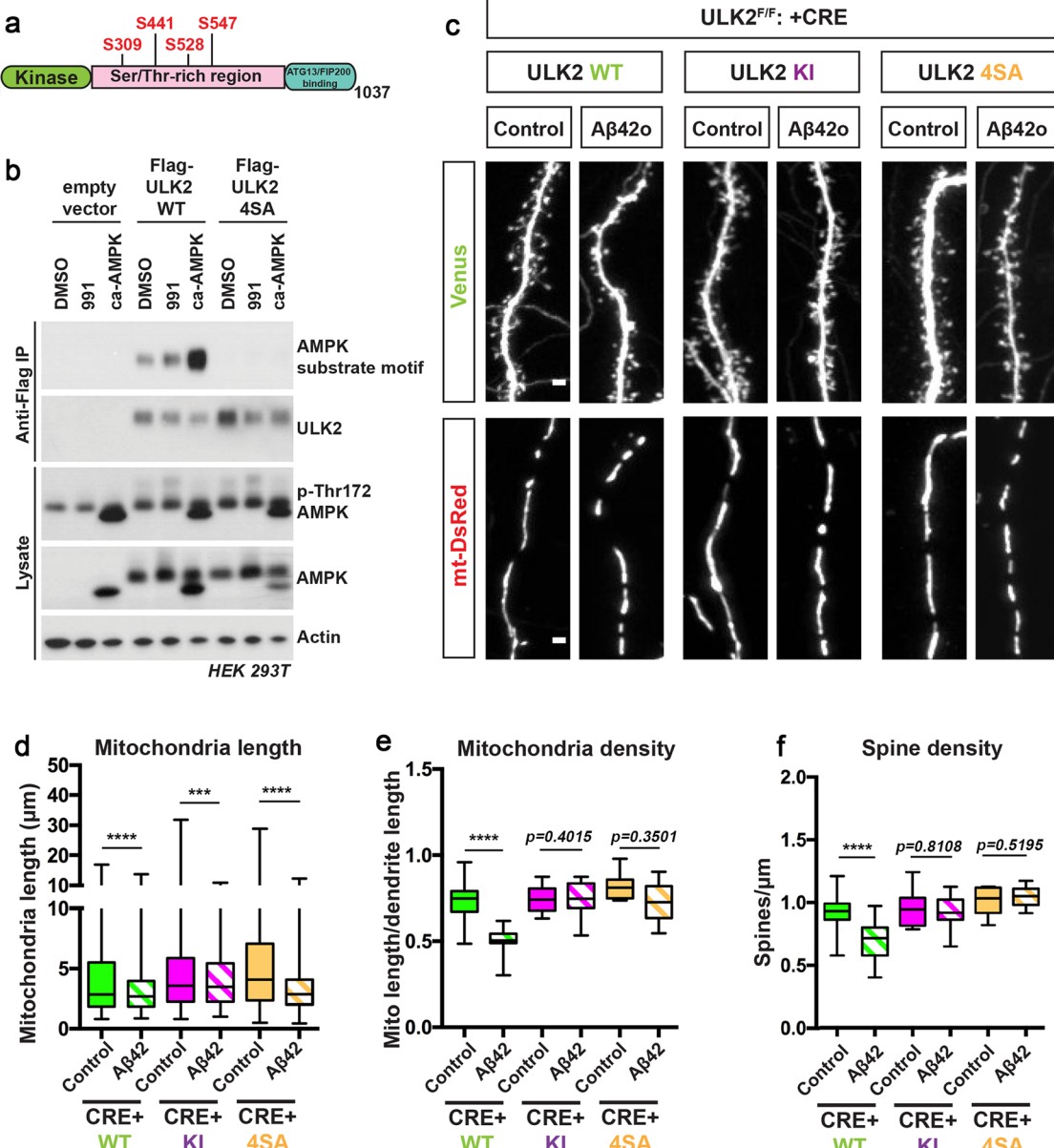

**Fig. 9 | AMPK-mediated phosphorylation of ULK2 is crucial for oligomeric Aβ42-induced synaptotoxicity and loss of mitochondrial biomass. a** Schematic of ULK2 domain structure highlighting the four predicted AMPK-mediated phosphorylation sites (see Supplementary Fig. S8) conserved in ULK1 and ULK2. **b** Flag-mULK2 WT or Flag-mULK2 4SA (the four conserved phosphorylation sites shown in Supplementary Fig. S8b are mutated to alanine) was overexpressed in HEK293T cells co-expressing either GST or GST-ca-AMPKα1(1–312) (constitutively active AMPK). Cells were treated with DMSO or 50 µM of compound 991 for 1 h. Blots were probed with AMPK substrate motif antibody, Flag M2 monoclonal antibody, total ULK2, p-Thr172 AMPK, total AMPK, and actin antibodies.
**c** Representative high-magnification images of secondary dendritic segments showing dendritic spines in the upper panel and mitochondria in the lower panel. ULK2$^{F/F}$ embryos were ex utero electroporated at E15.5 with pCAG-mVenus, pCAG-mito-DsRed, either a scrambled pCAG-Cre (Control, no Cre), or pCAG-Cre recombinase. The pCAG-Cre recombinase conditions in which ULK2 is genetically removed were also co-electroporated with either pCAG-mULK2 WT (Wild type),

pCAG-mULK2 KI (K39I) (Kinase inactive), or pCAG-mULK2 4SA (Kinase dead, see Supplementary Fig. S8). Neurons were treated at 20DIV with either a vehicle control or Aβ42o for 24 h. **d–f** Quantification of **d** dendritic mitochondrial length, (**e**) dendritic mitochondrial density, and **f** spine density. Analyses were done blind to the experimental conditions and done by manual counting using FIJI. For western blot in panel b, the same results were obtained for three independent experiments. In panels **d–f**, data are represented by box plots displaying minimum to maximum values, with the box denoting 25th, 50th (median), and 75th percentiles from three independent experiments. $n_{ULK2WT\ Control} = 32$ dendrites, 194 mitochondria; $n_{ULK2WT\ Aβ42o} = 30$ dendrites, 289 mitochondria; $n_{ULK2KI\ Control} = 36$ dendrites, 183 mitochondria; $n_{ULK2KI\ Aβ42o} = 34$ dendrites, 449 mitochondria; $n_{ULK24SA\ Control} = 34$ dendrites, 198 mitochondria; $n_{ULK24SA\ Aβ42o} = 32$ dendrites, 472 mitochondria. Statistical analyses were performed using a Mann–Whitney test in (**d–f**). Exact P values are indicated on the figure when available through Prism software, otherwise, the test significance is provided using the following criteria: *P < 0.001; ****P < 0.0001. Scale bar for magnified dendritic segments = 2 µm.

dendritic mitochondria is causally linked to excitatory synaptic loss was unknown, in part because the molecular mechanisms mediating Aβ42o-dependent mitochondrial remodeling were unknown.

AMPK is best-characterized as a metabolic sensor in nonneuronal cells[39]. Importantly, in AD patients, catalytically active AMPK is

abnormally accumulated in the cytoplasm of PNs of CA1, the entorhinal cortex, and the neocortex[37,38]. AMPK overactivation has been consistently reported in AD patient brain samples[37,38], in various AD mouse models[2,34,35], as well as in vitro upon acute treatment with Aβ42o[2,34,36,74].

The CAMKK2-AMPK signaling pathway is unique in its involvement in multiple phenotypes observed during the progression of AD. The present study and previous reports collectively implicate this signaling pathway as the first to unify multiple phenotypes reported in the brain of AD patients and in AD mouse models, including Ca²⁺ homeostasis disruption, APP processing, hTau phosphorylation, mitochondrial remodeling, increased autophagy, and synaptic loss[1,75,76]. There is growing evidence that one of the earliest events upon Aβ42o application is disrupted cytoplasmic Ca²⁺ homeostasis, which drives PNs hyperexcitability (see below)[29,77–81]. This increased Ca²⁺ accumulation precedes spine morphological changes[77], strongly suggesting that Ca²⁺-dependent signaling pathways, including the CAMKK2-AMPK pathway, occur early in the disease progression and play a critical role in reduced synaptic plasticity (reduced LTP-augmented LTD) through abnormal trafficking and post-translational modification of Ca²⁺ permeable AMPA receptors as well as NMDA receptors[66,82–85]. Importantly, recent work demonstrated that preventing AMPK-overactivation blocks the synaptic plasticity defects triggered by Aβ42o application through the ability of AMPK to regulate eukaryotic elongation factor 2 (eEF2) and its kinase's, eEF2K, control of protein translation[34].

Interestingly, Aβ42o-induced AMPK overactivation has been found to induce a vicious cycle of increased Aβ42o production via activation of JNK3-dependent APP processing[74]. In line with the results in ref. [74], we found that part of the protective effect of preventing Aβ42o-mediated AMPK overactivation might also be to reduce APP processing and Aβ42 production. However, the main results from our current study and previously published work is that AMPK links the two key players of AD: Aβ42o and Tau[2]. Catalytically active AMPK has been shown to co-localize with phosphorylated-S262 hTau in brains of AD patients[38], and previous work highlights that hyperactivation of AMPK by Aβ42o and/or other AMPK-like kinases such as NUAK1 leads to Tau phosphorylation on two of its serine residues (S262 and S356[2,86]) embedded in the R1 and R4 microtubule binding domains, respectively[87]. These results identified for the first time the CAMKK2-AMPK signaling pathway as a critical link between Aβ42o and Tau phosphorylation in the induction of synaptic loss[54].

In this paper, we identified two new critical effectors of AMPK mediating the effects of Aβ42o on synaptic loss. We show that AMPK overactivation by Aβ42o coordinates multiple aspects of mitochondrial remodeling by co-regulating mitochondrial fission and mitophagy through its activation of MFF and ULK2, respectively. For the first time, we demonstrate that Aβ42o mediated loss of excitatory synapses in cortical PNs in vitro and in CA1 PNs in vivo requires AMPK-mediated activation of MFF-ULK2-dependent mitochondrial fission and mitophagy, respectively. We cannot formally exclude the possibility that Aβ42o-mediated spine loss require ULK2-mediated autophagy events of cargoes other than mitochondria. However, this is unlikely since we also show that preventing MFF-dependent mitochondrial fission, which blocks subsequent loss of mitochondrial biomass (Figs. 5–7), also prevents Aβ42o-mediated spine loss in vitro and in vivo.

We have also identified that Aβ42o-mediated AMPK overactivation also signals through phosphorylation of Tau on S262 in conjunction with MFF-ULK2 phosphorylation/activation, leading to dendritic mitochondrial remodeling and synapse loss. Further work will be required to understand if Aβ42o-induced AMPK-Tau signaling plays a direct role in mitochondrial fission and/or mitophagy machinery, or if it plays an indirect role through its postsynaptic signaling function[67,88] and/or its ability to regulate microtubule dynamics[89].

Lastly, what makes this molecular mechanism attractive for the development of new therapeutic approaches is that CAMKK2-AMPK is a bona fide stress-response signaling pathway[54], i.e., these AMPK-dependent pathways are dispensable for normal neuronal development and/or synaptic maintenance in adult cortical and hippocampal neurons[54,90].

Mitochondrial remodeling and dysfunction are converging mechanisms of disease pathogenesis shared by multiple neurodegenerative diseases (ND). This is in part explained by the disruptions of mitochondrial homeostasis, motility, and dynamics observed in various NDs, as well as by genetic evidence suggesting a strong association between genes involved in mitochondrial function and various adult onset ND syndromes[88,91]. In AD specifically, altered mitochondrial dynamics have been implicated by changes of mitochondrial fission and fusion protein expression levels in AD patient brains[24] and in various AD mouse models[25], as well as altered post-translational modification of Drp1, which is thought to disrupt mitochondrial dynamics[24,71,73]. In line with our results, Wang and colleagues[24,73] also observe that, in the presence of Aβ42o, dendritic mitochondria decrease in length and density, which is correlated with reduced spine density. Interestingly, a recent study suggested a significant degree of mitochondrial structural remodeling in CA1 of the hippocampal region in various AD mouse models and in AD patients using 3D EM reconstructions[27]. Our present study further corroborates these observations and provides a molecular mechanism mediating Aβ42o-dependent mitochondrial remodeling and synaptic loss during the early stages of AD.

A second cellular defect thought to contribute to several neurodegenerative diseases, including AD, is a disruption of the autophagocytic–lysosomal pathway[41,76]. Although the cellular and molecular mechanisms of autophagy have largely been examined using nonneuronal cells[92], recent results demonstrated unique, compartment-specific regulation of autophagy in neurons. Local biogenesis of autophagosomes has been observed in distal axons, which are then trafficked retrogradely towards the soma in order to fuse with lysosomes, whereas autophagosomes in the soma are more stationary[93]. The dynamics of autophagy in dendrites, however, remain largely unexplored. Much of what we know about neuronal autophagy has been primarily focused on axonal and presynaptic autophagy dynamics, where it mediates efficient synaptic transmission by regulating the turnover of synaptic proteins[94,95]. Autophagy has received significant attention in AD research, in part due to the abnormal accumulation of autophagosomes and autolysosomes in dystrophic neurites of AD patients[76]. Moreover, familial mutations and polymorphisms associated with AD are linked to dysfunction of the autophagocytic–lysosomal pathway. Mutations of the Presenillin 1 and 2 proteins have been shown to disrupt lysosomal acidification and autophagy[96,97]. AD associated genes such as PICALM, BIN1, RAB11, and VPS34 are involved in various initiation, sorting, and trafficking processes of autophagy[98–101]. However, the majority of the cellular phenotypes involving autophagosome and autolysosome buildup in neurons have been described at late stages of AD progression when neurodegeneration involving neuronal cell loss occurs[76]. More recent results suggested that, in AD mouse models, autophagic flux is disrupted[35] and the expression level of mitophagy-associated genes are elevated during the early stages of AD[102].

A recent study confirmed that AMPK is overactivated in brains of AD human patients and is accompanied by increased MFF phosphorylation on S146 confirming our present observations[37]. However, the same study reported reduced ULK1 activation, increased mitochondria damage but reduced mitophagy[37]. One possibility to explain the potential differences between our results and the results of ref. 37 is that in their study, ULK1 expression/activation and mitophagy where assessed on whole brain tissue which includes neurons and non-neuronal cells, including astrocytes and microglial cells where mitophagy might be differentially altered than in neurons. Another potential difference between our present results and ref. 37 results is that our in vitro and in vivo results examine the influence of Aβ42o-mediated spine loss using acute (24 h) exposure in vitro or early stages of Aβ42o accumulation in the J20 model (3 months i.e., before amyloid plaque appearance) whereas the Fang et al. paper examine mostly

ULK1 and mitophagy events in "late" AD patients at a point where neuronal viability and therefore autophagosome clearance might be compromised[41]. Future experiments will need to address the timing and cell-type specificity of mitophagy activation or downregulation in both mouse models and human AD patients at early versus late stages of the disease progression.

Overall, our results demonstrate that Aβ42o triggers an AMPK-ULK2-dependent increase in mitophagy cell-autonomously in cortical and hippocampal mouse and human neurons, resulting in significant local degradation of dendritic mitochondria. We developed a direct way to visualize mitophagy using quantifications of mitochondria "degradation" in time-lapse analysis where genetically encoded mitochondrial markers disappeared in location where LAMP1 + lysosome and LC3+ autophagosome fused. We computed a "mitophagy index" which reflects mitochondria biomass loss at location where autophagolysosomes emerge, however, this index might also be influenced by local mitochondria dynamics, including fission events followed by mitochondria transport independent of mitophagy events. Despite this potential limitation, our rescue experiments in PNs performed by gene replacement strategies reinforces that MFF-dependent fission coupled with ULK2-dependent autophagy is causally linked with dendritic spine loss induced by Aβ42o. Altogether, our results strongly argue that at least during the early stages of the disease progression, Aβ42o-induced overactivation of CAMKK2-AMPK mediates synaptic loss through drastic structural remodeling and loss of biomass of dendritic mitochondria.

Our in vivo analyses also reveal a striking and previously unknown degree of compartmentalization of mitochondria morphology in dendrites of CA1 PNs: in basal and apical oblique dendrites, mitochondria are small and punctate, whereas the distal apical dendrites contain elongated and fused mitochondria (Supplementary Fig. S1). The transition between these two types of mitochondrial morphology is sharp and corresponds to the transition between two hippocampal layers, stratum radiatum (SR) and stratum lacunosum moleculare (SLM), where the earliest excitatory synaptic loss is observed in this mouse model. This is particularly relevant because the synapses made between medial entorhinal cortical inputs and the apical tufts of CA1 PNs in SLM has been described as one of the central synapses defective in late-onset forms of AD[103]. Future investigations will need to determine why this synapse is particularly vulnerable during early stages of AD progression.

As discussed above, disruption of Ca$^{2+}$ homeostasis has been observed in various AD mouse models[29,78,80,104], in wild-type PNs exposed to acute treatment of oligomeric Aβ42 in vivo[77], and neuronal cultures exposed to Aβ42o in vitro[105,106]. Recent results suggested that disruption of Ca$^{2+}$ dynamics is largely due to Aβ42o, as immune-depletion of Aβ42o can prevent this phenotype[77] and decrease the population of hyperactive neurons[29,78]. Possible explanations of this increased cytoplasmic Ca$^{2+}$ accumulation include increased mobilization of Ca$^{2+}$ from the extracellular space through NMDAR and VGCC, increased leakage of Ca$^{2+}$ from intracellular Ca$^{2+}$ storing organelles such as the ER, and loss of Ca$^{2+}$ buffering capacity by the neurons. Currently, there is evidence supporting the hypothesis that excess Ca$^{2+}$ is derived from the extracellular influx of Ca$^{2+}$ via NMDA Receptors[1,77], as well as from the ER via the IP$_3$ and Ryanodine Receptors[107,108]. Future experiments will need to address the exact source of intracellular Ca$^{2+}$ increase (NMDA/VGCC only and/or ER-mitochondria dependent intracellular stores/buffering) leading to CAMKK2 overactivation by Aβ42o. The fact that ER-mitochondria contacts have been shown to be disrupted in response to Aβ42o (ref. 109 and reviewed in ref. 110) argues in favor of disruption of ER-dependent Ca$^{2+}$ or reduced mitochondrial Ca$^{2+}$ buffering. Regardless of the Ca$^{2+}$ source, the current model is that PNs ultimately experience an increase in frequency and/or amplitude of intracellular Ca$^{2+}$ that can activate Ca$^{2+}$-dependent signaling pathways, including overactivation of the CAMKK2-AMPK

pathway. There may be a complex, positive feed-forward loop where Aβ42o-dependent, fast NMDAR-dependent Ca$^{2+}$ influx[77] overactivates the CAMKK2-AMPK pathway[2], resulting in the reduction of mitochondrial biomass, i.e., mitochondrial matrix volume, which in turn reduces Ca$^{2+}$ buffering capacity, further increasing cytoplasmic Ca$^{2+}$ accumulation and CAMKK2 overactivation. Future work will need to address if the striking but spatially restricted degree of Aβ42o-dependent reduction in dendritic mitochondrial biomass and subsequent reduction in spine density identified in the present study in the apical tuft dendrites of CA1 PNs in the J20 AD mouse model contributes to the degradation of the spatial tuning properties of these neurons at early stages of the disease progression.

## Methods

### Animals
Mice were used according to protocols approved by the Institutional Animal Care and Use Committee (IACUC) at Columbia University and in accordance with National Institutes of Health guidelines. Time-pregnant CD1 females were purchased from Charles Rivers. 129/SvJ, C57Bl/6J nontransgenic mice, and hemizygous transgenic mice from line B6.Cg-*Zbtb20*$^{Tg(PDGFB-APPSwInd)20Lms}$/2Mmjax (hereafter referred as J20) (The Jackson Laboratory stock #006293) were maintained in a 12-hour light/dark cycle. J20 mice express human APP carrying the Swedish and Indiana mutations under a PDGFβ promoter[8,111]. AMPKα1$^{F/F}$α2$^{F/F}$ double conditional knockout mice were generated by Drs. Foretz and Viollet[50]. Ulk2$^{F/F}$ mice were obtained from The Jackson Laboratory stock #017977[60]. Timed-pregnant females were obtained by overnight breeding with males of the same strain. Noon the day after the breeding was considered as E0.5. Mice of both sexes were used for each experiment and no sex difference was detected.

All mouse strains were maintained in a 12 h dark/12 h light cycle in rooms with constant humidity and temperature control in a barrier SPF housing facility.

### Synthetic Aβ42 oligomers preparation
Aβ42 (rPeptide or Creative Peptide) was processed to generate Aβ42 oligomers as described in ref. 2. Briefly, lyophilized Aβ42 peptide was dissolved in hexafluoro-2-propanol (HFIP; Sigma-Aldrich) for 2 h to allow monomerization. HFIP was removed by speed vacuum, and the monomers were stored in −80 °C. Aβ42 monomers were dissolved in anhydrous dimethyl sulfoxide (DMSO) to make a 5 mM solution, then added to cold phenol red-free F12 medium (Life Technology) to make a 100 μM solution. The solution was incubated at 4 °C for 2 days and centrifuged at 12,000 × *g* for 10 min at 4 °C in order to discard the fibrils. The supernatant containing the 4 °C oligomers were assayed for protein content using the BCA kit (Thermo Fisher Scientific). For control experiments, vehicle treatment corresponded to the same volume of DMSO and F12 media used for Aβ42o treatment, and for Supplementary Fig. S02, a peptide corresponding to the inverted sequence of Aβ42 (INV42; rPeptide and Creative Peptide) was processed as for Aβ42 oligomerization. For western blotting of Aβ42 oligomers, 16.5% Tris-Tricine SDS-PAGE was performed. Synthetic Aβ42 oligomers were prepared in 2X Tricine sample buffer (BIORAD) without reducing agent and resolved by SDS-PAGE. The separated proteins were transferred onto Immobilin-FL PVDF membrane (EMD Millipore), blocked for 1 h with Odyssey blocking buffer (LICOR) and probed with 6E10 monoclonal antibody (Covance) overnight at 4 °C. Secondary antibody incubation was performed for 1 h at room temperature with IRDye 680RD goat anti-mouse secondary (LICOR) and the blot was visualized using the Odyssey Imaging system. To estimate the relative concentration of synthetic Aβ42 peptide monomer versus oligomers (dimers and trimers), we performed quantitative western blotting (see Supplementary Fig. S2e) and measured the ratio of optical density measured for the monomeric peptide versus the dimer/trimer bands. Using this approach, we estimated that a 1 μM concentration of

oligomerized peptide contains ~300–450 nM of effective dimers/trimers (Supplementary Fig. S2e).

## Primary cortical culture and ex utero electroporation

Cortices from E15.5 mouse embryos were dissected followed by dissociation in complete Hank's balanced salt solution (cHBSS) containing papain (Worthington) and DNase I (100ug/mL, Sigma) for 15 min at 37 °C, washed three times, and manually triturated in DNase I (100ug/mL) containing neurobasal medium (Life Technology) supplemented with B27 (1x, Thermo Fischer Scientific,), FBS (2.5%, Gibson), N2 (1×, Thermo Fischer Scientific), and glutaMAX (2 mM, Gibco). Cells were plated at $10.0 \times 10^4$ cells per 35 mm glass bottom dish (Mattek) that has been coated with poly-D-lysine (1 mg/mL, Sigma) overnight. One-third of the medium was changed every 5 days thereafter with non-FBS-containing medium and maintained for 20–25 days in 5% $CO_2$ incubator at 37 °C. Ex utero electroporation was performed as previously published[112]. See the Supplemental Experimental Procedures for details and constructs. Plasmids used for ex utero electroporation where all in pCAG vector backbone[113] expressing the following cDNAs: LAMP1-mEmerald[114], mito-DsRed and Venus[112], mRFP-LC3 (originally from https://www.addgene.org/21075/ and subcloned into pCAG backbone). pSCV2-Venus and pCAG-mito-dsRed were described in a previous publication[114]. pLKO-MFFsh specific for mouse was obtained from Sigma-Aldrich (TRCN0000174665). Validation of the MFF shRNA knockdown efficiency was done elsewhere[14]. cDNA encoding Renalin, MFF WT (Uniprot Q9GZY8 isoform 5), and MFF-AA were described in ref. 40 and subcloned in the pCAG plasmid backbone 3′ to the CAG promoter using PCR. 0.25 μg/mL of individual constructs were used for the MFF rescue experiments. pCAG-Cre, pCAG-Scrambled Cre, pCIG-hTau (isoform 4R2N), and non-phosphorylatable hTau-S262A were described in previous publications[112]. pcDNA3-flag-ULK2 specific for mouse was obtained from Addgene (#27637) and was used to generate the kinase-inactive (K39I) and serine-to-alanine 4SA mutant (S309/T441/S528/S547A) using Quickchange II site-directed mutagenesis kit (Agilent Technologies) and were subcloned into pCAG vector backbone using PCR. pSUPER-scrambled, ULK1 shRNA and ULK2 shRNA constructs were validated in a previous study[115]. Sequences for ULK1 shRNA: 5′-ATCCCCAGACTCCTGTGACACAGATTTCAAGAGAATCTGTGTCACAGGAGTCTTTTTTA-3′ and for ULK2 shRNA: 5′-ATCCCCTGCCTAGTATTCCCAGAGATTCAAGAGATCTCTGGGAATACTAGGCATTTTTA-3′.

## In utero hippocampal electroporation

In utero electroporation targeting the hippocampus was performed as previously described[2,116,117] with slight modifications as described in refs. 55, 118 to target the embryonic hippocampus at E15.5. See Supplemental Experimental Procedures for more details. In brief, a mix of endotoxin-free plasmid preparation was injected into both lateral hemispheres of E15.5 embryos using a picospritzer. Electroporation was performed with a triple electrode to target hippocampal progenitors in E15.5 embryos by placing the two anodes (positively charged) on either side of the head and the cathode on top of the head at a 0° angle to the horizontal plane. Four pulses of 45 V for 50 ms with 500-ms interval were used for the electroporation.

## Microscopy

Imaging on dissociated neurons was performed between 20–25 DIV in 1024 × 1024 resolution with a Nikon Ti-E microscope equipped with A1R laser-scanning confocal microscope using the Nikon software NIS-Element (Version 4.3; Nikon, Melville, NY, USA). We used the following objective lenses (Nikon): 10× PlanApo; NA 0.45 (for images of hippocampal slices), 60× Apo TIRF; NA 1.49 (for analysis of spine density and mitochondrial morphology in cultured neurons), 100× H-TIRF; and NA 1.49 (for analysis of spine densities and mitochondrial morphology in brain slices). Dendritic spine density

was quantified on secondary or tertiary dendritic branches that were proximal to the cell body, on z projections for cultured neurons and in the depth of the z stack for slices, using FIJI software (ImageJ; NIH). See Supplemental Experimental Procedures for more details.

## hESC culture

H9 (WA09; WiCell) and its genome-edited derivatives were maintained on StemFlex (Life) and Cultrex substrate (Biotechne), and routinely split approximately twice a week with ReLeSR (Stem Cell Technologies) in the presence of ROCKi (Y-27632; Selleckchem). H9 is a commercially available hESC line on the NIH Registry (# 0062), and Dr. Sproul has approval to genome edit and differentiate this hESC line by the Columbia University Human Embryonic and Human Embryonic Stem Cell Research Committee.

## APP$^{Swe}$ knockin hESC line generation

The APP$^{Swe}$ mutation (KM670/671NL) was knocked into both alleles of H9 using CRISPR/Cas9, bi-allelic knockin of APP$^{Swe}$ has been demonstrated to have increased Aβ42 production relatively to monoallelic knockin in a control iPSC line[48]. In brief, a sgRNA targeting Exon16 of *APP* was designed and subcloned into the MLM3636 vector, a gift from Keith Joung (Addgene plasmid # 43860; http://www.addgene.com/43860). An ssDNA HDR template (IDT) was designed in which the APP$^{Swe}$ mutation disrupted the PAM and introduced a de novo Xba1 restriction site. The sgRNA, Cas9-GFP (a gift from Kiran Musunuru (Addgene plasmid # 44719; http://www.addgene.org/44719/), and ssDNA HDR template were electroporated (Lonza nucleofector) into feeder-free H9 hESCs, followed by cell sorting on GFP signal and plating at low density on MEFs (MTI-GlobalStem). Individual clones were manually picked into a 96 well format, and subsequently split into duplicate plates, one of which were used to generate sgDNA as had been done previously[48]. For each clone, exon 16 of *APP* was amplified and initially screened by restriction digest with Xba1 (NEB). Sanger sequencing was used to confirm the mutation, and successful knockin clones were expanded and banked. Potential off-target effects of CRISPR/Cas9 cleavage were analyzed by Sanger sequencing of the top 5 predicted off-target genomic locations [https://mit.crispr.edu], which demonstrated a lack of indels for multiple clones. One APP$^{Swe}$/APP$^{Swe}$ knockin line was analyzed and demonstrated to have a normal karyotype (G-banding, Columbia University Clinical Cytology Laboratory), and was used in the present work (Cl. 160).

## Transdifferentiation of hESCs into cortical-like pyramidal neurons

pLV-TetO-hNGN2-eGFP-Puro was generously provided by Kristen Brennand[119] and FUdeltaGW-rtTA was a gift from Konrad Hochedlinger (Addgene plasmid #19780; http://www.addgene.org/19780/)[120]. Concentrated lentiviruses for these two plasmids were made using Lenti-X concentrator (Takara) as has been done previously[121]. Permanent doxycycline-inducible Ngn2-eGFP lines were generated by reverse infecting H9 and H9:APP$^{Swe/Swe}$ hESC lines with Ngn2-eGFP and rtTA and subsequent selection by puromycin.

For transdifferentiation of these two lines of hESCs into cortical neurons (iNs[49]), H9 and H9:APP$^{Swe/Swe}$ lines were plated at ~165,000 cells per 12-well in N2/B27 medium on PEI (0.1% Sigma)/laminin (10 μg/mL, Biotechne) coated plates, supplemented with 1 μg/mL doxycycline (Sigma), 10 ng/mL BDNF and NT3 (Biotechne), and 1 μg/mL laminin (Biotechne). BDNF, NT3, and laminin were added at similar levels for all subsequent feeds for the duration of the experiment. One day post plating, cells were treated with puromycin (1 μg/mL, Fisher). The following day, cells were treated with 2 μM AraC (Biotechne) for 4 days, and 100 ng/mL on subsequent feeds. One week post infection, cells were transitioned into BrainPhys (Stem Cell Technologies). Two weeks post infection, 10% mouse astrocyte conditioned media (ACM;

ScienCell) was also added to the medium. Transfection of mitoDsRed2 was done at day 21–23 post infection using Lipofectamine 3000 (Life), and cells were fixed and analyzed 4 days later. A full media change (1/2 saved conditioned media) was performed approximately 6 h post transfection. Note that iNs had poor survival from Lipofectamine transfection without either ACM or primary mouse glia present.

## Cell lysis and immunoprecipitation

Neuron and differentiated hESC cultures were washed with cold PBS and lysed in Triton Lysis Buffer: 20 mM HEPES (pH 7.5), 150 mM NaCl, 1 mM EDTA, 1 mM EGTA, 1% Triton X-100, 0.25 M Sucrose, Complete™ protease inhibitor (Roche), and phosphoSTOP (Roche). Lysates were incubated at 4 °C for 15 min and cleared at 15,000 × g for 15 min at 4 °C. Total protein was normalized using Bio-rad Protein Assay Dye (Bio-rad).

For western blotting of phospho-T172 AMPK and total AMPK, equal amounts of lysates were loaded on a Mini-Protean TGX (4–20%) SDS-PAGE (Bio-rad). The separated proteins were transferred onto polyvinylidene difluoride membrane (PVDF, Bio-Rad), blocked for 1 h with blocking buffer containing 5% fat-free dry milk in TBS-T. Membranes were then incubated overnight at 4 °C with different primary antibodies diluted in the same blocking buffer. Incubations with HRP-conjugated secondary antibodies were performed for 1 h at room temperature, and visualization was performed by chemiluminescence.

For phospho-T172 AMPK and total AMPK for eSC experiments, equal amounts of lysates (50 μg) were immunoprecipitated with phospho-T172 AMPKα or total AMPKα to enhance AMPK proteins. Lysates from H9-Control and H9-APP lines were incubated overnight at 4 °C in a rotator with either phospho-T172 AMPK or total AMPK primary antibodies. Protein A-Agarose beads (Sigma) were washed three times with PBS and added to the immuno-bound lysates for at least 1 h at 4 °C. The lysates and beads were then washed thrice with lysis buffer and eluted in SDS lysis buffer for 5 min at 95 °C and resolved by loading equal amounts of eluted protein on a Mini-Protean TGX (4–20%) SDS-PAGE (Bio-rad). The separated proteins were transferred onto polyvinylidene difluoride membrane (PVDF, Bio-Rad), and blocked for 1 h with Odyssey Blocking Buffer (PBS). Membranes were then incubated overnight at 4 °C with phospho-T172 AMPK or total AMPK primary antibodies diluted in the same blocking buffer. Incubations with Li-Cor fluorescence-coupled secondary antibodies were performed for 1 h at room temperature, and visualization was performed by Li-Cor Odyssey Blot Imager.

For phospho-MFF and MFF immunodetection, equal amounts of lysates (80 μg) were immunoprecipitated with MFF or HA (as negative control for specific binding) to enhance MFF proteins. SureBeads™ Protein A Magnetic Beads were washed three times with lysis buffer and added to equilibrated lysates for 1 h at 4 °C. The beads were washed two times with lysis buffer, two times with lysis buffer without Triton X-100, and then eluted by boiling in SDS lysis buffer for 5 min at 95 °C and resolved by SDS-PAGE (Bio-rad) gel. The separated proteins were transferred onto polyvinylidene difluoride membrane (PVDF, Bio-Rad), blocked for 1 h with blocking buffer containing 5% BSA in Tris-buffered saline solution. Membranes were incubated overnight at 4 °C with either phospho or total MFF primary antibodies diluted in the same blocking buffer. Clean-Blot™ IP Detection Kit (Thermo Fisher Scientific) was used to detect the bands of interest without the detection of denatured IgG as MFF protein size is close to both heavy and light chain. After overnight primary antibody incubation, membranes were incubated with Dilute Clean-Blot IP Detection Reagent at 1:100 for 1 h at room temperature and visualization was performed by enhanced chemiluminescence.

For phospho-ULK2 experiments, HEK293T cells (ATCC #CRL-3216) were transfected using Lipofectamine 2000 (Life Technologies) according to manufacturer's instructions with pcDNA3 plasmids encoding flag-tagged WT ULK2, flag-tagged K39I ULK2 (ULK2-KI) or flag-tagged S309A, T441A, S528A, S547A ULK2 (SA4). Where indicated, cells were co-transfected with a constitutively active truncated version of AMPKα1 (ca-AMPK). Transfected cell lysates were harvested 40–48 h post transfection. One hour after changing the media with fresh media containing vehicle or 50 μM compound 991 (Glixx laboratories, Inc.), cells were washed with cold PBS and lysed in lysis buffer: 20 mM Tris pH 7.5, 150 mM NaCl, 1 mM EDTA, 1 mM EGTA, 1% Triton X-100, 2.5 mM pyrophosphate, 50 mM NaF, 5 mM β-glycerophosphate, 50 nM calyculin A, 1 mM Na3VO4, and protease inhibitors (Roche). Lysates were incubated at 4 °C for 15 min and cleared at 16,000 g for 15 min at 4 °C. Total protein was normalized using BCA protein kit (Pierce) and lysates were resolved on SDS-PAGE gel. Immunoprecipitations were performed by adding magnetic M2 FLAG beads (Sigma) to equilibrated lysates for 2 h at 4 °C. The beads were washed three times with lysis buffer and then eluted by boiling in SDS lysis buffer for 5 min and resolved by SDS-PAGE.

Antibodies used in this study: anti-phospho-T172 AMPKα (40H9, 1:1000, Cell Signaling); anti-AMPKα1/2 (1:1000, Cell Signaling Technology); MFF (1:1000, Proteintech); phospho-S146/172-MFF (1:1000, CST); pAMPK motif (Cell Signaling Technology, #5759 S); total AMPKα (Cell Signaling Technology, #2532), p62/SQSTM1 (Cell Signaling #5114); Flag M2 (Sigma, F7425), pAMPK (Cell Signaling Technology, #2535), actin (Sigma, A5441). HRP-coupled secondary antibodies to mouse (AP124P) or rabbit (AP132P) were from Millipore. Li-Cor Fluorescence-coupled secondary antibodies to mouse or rabbit were from Li-Cor Biosciences.

## Tissue slice preparation and immunofluorescence

Animals were sacrificed 3 months after birth by terminal transcardial perfusion of 4% paraformaldehyde (PFA, Electron Microscopy Sciences) followed by overnight post-fixation in 4% PFA. Brains were washed 3X with PBS for 5 min and sectioned at 100-μm thickness using vibratome (Leica). Sections were permeabilized with 0.2% Triton X-100 3X for 5 min, blocked in PBS-based blocking buffer with 5% BSA and 0.2% Triton, and stained with chicken anti-GFP (Aves, 1:1000) and rabbit anti-RFP (Abcam, 1:1000) for enhancement of PSCV2-mVenus and pCAG-mito-DsRed, respectively overnight at 4 °C. Sections were washed with 0.2% Triton buffer 3X for 5 min and incubated with Alexa488- and Alexa 555- labeled secondary antibodies for PSCV2-mVenus and pCAG-mito-DsRed, respectively overnight at 4 °C. Sections were washed with 0.2% Triton buffer 3X for 10 min each and mounted using VectaShield® Mounting Medium (Vector Laboratory).

## Drug treatments

Neuronal cells at 20–25 DIV were treated with STO-609 (2.5 μM, Millipore), a CAMKK2 inhibitor, 2.5 h prior to Aβ42 treatment. N2A cells were transfected with pCAG-MFF-WT construct using the recommended protocol from FuGENE®HD Transfection Reagent overnight. Cells were treated with either DMSO or Metformin (2 mM, Sigma) for 5 h before cells were lysed using Triton lysis buffer as described under Cell Lysis and Immunoprecipitation.

## Analysis of spine density and mitochondrial morphology

Dendritic spine densities were quantified on secondary or tertiary dendritic branches for cultured neurons and in the depth of the z stack for slices, using FIJI software (ImageJ, NIH). The length of the dendritic segment was measured on the z projection, which implies the density could be overestimated. To limit this issue, only dendrites that were parallel to the plane of the slices were analyzed. Spines were quantified over an average of 40μm in cultures (between 2 and 3 segments per cell) and 40μm in vivo (between 2 and 3 segments per cell). Spine density was defined as the number of quantified spines divided by the length over which the spines were quantified. The criteria for

measuring mitochondrial length for each dendritic segment were the same as above. Mitochondrial density was defined as the total sum of all the mitochondria divided by the length over which the mitochondria lengths were measured.

## Analysis of mitophagy events

For quantifications in Fig. 3, manual counting of the total number of dendrites and number of dendrites showing accumulation of fluorescently-tagged LC3-RFP and LAMP1-mEmerald was performed for all imaged neurons. To confirm both LC3 and LAMP1 accumulation over time following Aβ42o application using an independent quantification approach, we selectively examined all of the 251 dendritic segments categorized as showing LC3+/LAMP1+ accumulation upon Aβ42o application and measured the intensities of LC3 and LAMP1 fluorescence (within an ROI ranging from 20–30μm in length) over time. All the dendrites that showed accumulation of LC3 and LAMP1 in control conditions were also analyzed, as well as randomly selected dendritic segments that did not show detectable accumulation of LC3 and LAMP1. Optical density of LAMP1-mEmerald and LC3-RFP fluorescence were normalized to the initial starting fluorescence intensities at $t_0$ ($\Delta F/F_0$). For these selected dendrites, mito-mTagBFP2 intensities were also measured over time for Fig. 3e. To measure local mitophagy events as indicated in Fig. 3d, random dendritic segments were selected for each condition for analysis as shown in Fig. 3a. For a given dendritic segment, intensity line scan analysis (pixel neighborhood-1 pixel) was performed where two vertical lines were drawn: one along the region of dendrite without LC3/LAMP1 puncta and one along the region with LC3/LAMP1 puncta. The fold change of mitochondrial intensity from time point zero to 14 h after respective treatment was calculated as illustrated in Fig. 3f.

## Analysis of Mitochondrial Density in hES cell-derived cortical neurons

For Supplementary Fig. S3B, C, lines of equal thickness and of an average length of ~40 μm were drawn along the neurites of differentiated hES cell-derived cortical neurons in Nikon NIS-Elements Software. A kymograph was subsequently created from each drawn line, for which the mean fluorescence intensity of the mitochondria as a function of the area was calculated and normalized to the background fluorescence intensity of the image.

## Statistics

Statistical analyses were performed with Prism 9.0.4 (GraphPad Software, LLC). The statistical tests applied for data analysis are indicated in the corresponding figure legend. Experimental groups with Gaussian/normal distributions were assessed using the unpaired $t$ test for two-population comparison. For groups that deviated significantly from normal distributions, a nonparametric test (Mann–Whitney $U$ test for two-population comparisons) was used. Unless otherwise noted, data are expressed as mean ± SEM. For dendritic spine and mitochondrial morphology analysis, all data were obtained from at least three independent experiments or at least three individual mice of each genotype. The test was considered significant when $P < 0.05$. For all analyses, the following apply: *$P < 0.05$; **$P < 0.01$; ***$P < 0.001$; ****$P < 0.0001$; ns, not significant with $P > 0.05$.

## Reporting summary

Further information on research design is available in the Nature Research Reporting Summary linked to this article.

## Data availability

The datasets generated during and/or analyzed during the current study are available from the corresponding author upon reasonable request. Source data are provided with this paper.

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

## Acknowledgements

We thank members of the Polleux lab for feedback and discussion, as well as Karen Duff, Ulrich Hengst, and Carol Troy for critically evaluating the manuscript. We thank Fan Wang (Duke University) for providing ULK1 and ULK2 shRNA constructs. We thank Qiaolian Lu and Miyako Hirabayashi for the excellent management of our mouse colony. This work was supported by grants from the NIH (NS089456) (F.P.), a F31 Award from NIH (A.L.), a K99 award (NS091526) (T.L.L.), the Thompson Foundation (TAME-AD) (A.S.), the Henry and Marilyn Taub Foundation (A.S.), the Ludwig Foundation (F.P., A.S., and A.M.V.), and an award from the Fondation Roger De Sproelberch (F.P.). Georges Mairet-Coello is affiliated with UCB Biopharma (Belgium) but has no other competing interests to declare.

## Author contributions

A.S., C.K., D.M.V., T.L.L. R.S., and F.P. contributed to the conceptualization of this manuscript. A.L., C.K., D.M.V., S.H., A.S., and F.P. contributed to the methodology of this manuscript. A.S., C.K., D.M.V., T.L.L., G.M-C., S.Y.K., A.A., M.F., B.V., and A.S. contributed to the investigation presented in this manuscript. A.S., C.K., D.M.V., and F.P. participated in the writing of this manuscript. A.S., R.S., and F.P. secured funding for the work presented in this manuscript. A.S., R.S., and B.V. provided resources which enabled the work presented in this manuscript.

## Competing interests

The authors declare no competing interests.
