## [Peer Review File · Nature Communications]

Reviewers' Comments:

Reviewer #2:

Remarks to the Author:

The manuscript by Lee and colleagues builds on previous work from the Polleux group and others to further investigate the pathway leading to mitochondrial alteration and subsequent spine loss. It has previously been illustrated that pathogens, such as AB42o alters AMPK activity and results in mitochondrial fragmentation and abnormal mitochondrial distribution (Thornton et al 2011, Yoon et al 2012, Mairet-Coello et al 2013). In the present study the authors firstly confirm the previous findings and build on this to identify the specific mechanism leading to mitochondrial fission (MFF) and mitophagy (ULK2) which results in spine loss.

While this advance is modest, by uncovering the full molecular pathway as it relates to the mitochondria, it does uncover potential translational findings; the ability to inhibit the pathways (MFF and ULK2), thereby inhibiting spine loss without altering normal neuronal development or synaptic maintenance. Therefore, opening in possibility as a potential early stage therapeutic target. However, the paper does not show how the mitochondria alterations result in spine loss, merely that they are required for the spine loss to occur. Furthermore, the current manuscript has a limited scope of mechanistic understanding compared with authors previous publication.

1. The authors uncover an interesting finding in that the hippocampal CA1 neurons present with a compartmentalised mitochondrial distribution in comparison to the cortical neurons. It would be beneficial to the manuscript if the authors had investigated this difference e.g. do the hippocampal mitochondria respond the same way to pathogens as the cortical neurons, or do they show different responses based on the compartment.

This is potentially illustrated, but not investigated/expanded, in figure 2. In which Abeta treatment does not alter the mitochondria length or density in the apical oblique (CA2/CA3) or basal compartments. This is surprising as previous research has shown reduced LTP upon Abeta incubation which requires schaffer collateral pathway (CA3 – CA1). The authors should explain how the findings of the manuscript fit into the literature regarding this point, and/or complete experiments at older ages.

2. The authors state that the PNs of CA1 show compartmentalisation while cortex does not, while this has been shown for juvenile animals (p21 – figure 1) it was not shown for adult animals (p90 – figure 2). Therefore, the authors should clarify in the text the age ranges etc or add adult cortical data.

3. While the authors show spine density reductions it is becoming more apparent that spine type (morphology) is important. Can the authors examine the morphology of the spines, particularly in the hippocampal data, spine changes may have been missed in the apical oblique region by only examining density.

4. Was spine loss associated with mitochondria changes? – e.g. Did a larger alteration in mitochondria result in greater spine loss (time lapse imaging required)? Or are the mitochondria changes uniform across the whole of the cortical neuron.

5. In main text line 371, authors discussed “disrupted cytoplasmic Ca²⁺ homeostasis that drives PNs hyperexcitability.....”. It would be better to include and discuss Ca²⁺-permeable AMPA receptor hypothesis in Abeta-mediated synapse dysfunction. There is a literature described rapid aberrant calcium permeable AMPA receptor hypothesis (Whitcomb et al, 2015?). This would be one of others which could contribute mitochondria-mediated synapse dysfunction.

Other points:

In FIG2.D Mitochondria look longer and brighter in this figure

FIG3 A'-C' I can see the reduction of mitochondria in terms of number of elements but they seem brighter, explain this why?

In FIGURE 4 B and C are inverted

In FIGURE 7 all the figure legends the letters indicating the relative images are a mix of capital and small size and not always proper matched.

Reviewer #3:

Remarks to the Author:

In the current manuscript, Lee et al. demonstrated that MFF and ULK2 mediate amyloid-beta-induced structural changes, reflected by reduction in dendritic mitochondrial length/density and synapse loss. These molecular pathways were linked to AMPK-CaMKK2, previously described by the authors to mediate amyloid-beta-induced synapse loss.

The mechanism of synaptic loss is a long-standing question in AD biology. Accumulated evidence suggests that ER-Ca plays a crucial role in this process and many molecular players have been already discovered. The authors stated "For the first time, we demonstrate that A β 42₀ mediates loss of excitatory synapses through MFF-ULK2-dependent mitochondrial fission and mitophagy". Unfortunately, they did not provide the mechanisms linking MFF and ULK2 to synaptic loss. Mitochondrial fragmentation / reduction in mitochondrial mass, well known AD phenotypes, do not explain why synapses become unstable and disappear. Clearly, the structural mitochondrial changes are associated with functional mitochondrial deficits - such as energy / calcium / redox imbalance - that remain to be identified as a potential cause of synapse loss.

MAJOR:

1. The authors previously showed (Lewis et al, 2018) that MFF knockdown (shMFF) causes a reduction in mitochondrial length exclusively in axons, but not in dendrites in WT neurons. In the same work, shMFF was shown to increase Ca buffering capacity of axonal mitochondria, inhibiting cytosolic Ca transients and synaptic vesicle release. In the submitted manuscript, shMFF occluded amyloid-induced reduction in mitochondrial length and density in dendrites, without having effects on its own in the dendritic compartment. Thus, the main question is how amyloid-beta converts dendritic mitochondria from being MFF-independent to MFF-dependent?

There are several possibilities. First, amyloid-beta may affect putative signaling pathways connecting MFF to mitochondrial fragmentation. The second possibility is that alteration of presynaptic function by amyloid-beta precedes synapse loss and these presynaptic alterations constitute the major cause of synapse loss. Therefore, the effects of amyloid-beta on presynaptic functions (Ca and release of synaptic vesicles) and the effects of shMFF on these amyloid-beta-induced presynaptic changes should be tested directly. Previous study shows deficits in density, trafficking and functioning of axonal mitochondria by amyloid-beta (Du et al, PNAS 2010) which may impact synapse loss in vivo.

It would be important to test several time points, including those preceding synapse loss. The effect of amyloid-beta on cytosolic and axonal mitochondrial Ca, axonal mitochondrial length/density, as well as on synaptic vesicle release should be quantified. Based on the literature, amyloid-beta may cause a wide range of presynaptic effects, from augmentation to inhibition, depending on its biophysical properties (that's why it would be preferable to make knockdown in J20 animals). The authors should test whether shMFF occludes presynaptic effects of amyloid-beta under the experimental conditions used in the study. Reduction in the number of spines could be secondary to inactivity of presynaptic terminals or a compensatory mechanism to hyperactivity (presynaptic or hyperexcitability).

In the same line, AMPK has been shown to regulate presynaptic function in other brain regions (Yang et al., Cell 2011).

2. Does the effect of shMFF on mitochondrial length depend on ER-Ca? This is a critical question to answer in order to connect the present study to numerous previous findings in the field (the authors simply ignored all of them despite working on Ca-dependent CaMKK2-AMPK pathway). Does amyloid-beta cause ER-Ca overload that leads to overload of mito-Ca? Does shMFF rescue mito-Ca overload? Recent work suggests that amyloid-beta modulates ER-mito Ca crosstalk in

hippocampal neurons and this effect is age-dependent (Calvo-Rodriguez et al., 2019). Misregulation of ER-mito Ca crosstalk by amyloid-beta may be reason of synapse loss. What is the role of PDZD8, discovered by the authors, in this mechanism?

3. The authors suggest that mitochondrial fragmentation causes spine loss, while the opposite may be true. For this, they have to demonstrate the time cause of fragmentation vs time-cause of synapse loss in vitro and in vivo.

4. An alternative possibility is that MFF/ULK2/AMPK affect amyloid-beta levels in J20 mice and this is the main reason for spine loss in vivo. The authors should check how knockdown of these proteins affects amyloid-beta levels and amyloid-beta 42/40 ratio and whether it rescues synapse loss in vivo. Artificial addition of a mixture of monomeric and oligomeric amyloid-beta 42 preparation that causes synapse loss in 24 hr in cultures is very different from slow process of synapse loss in vivo that starts at distal dendrites in 3-m.o. J20 mice, highly likely are evident at many other dendrites at later stages.

5. The article begins from showing that in vivo mitochondria morphology in cortical L 2/3 and in CA1 apical tuft dendrites are mostly elongated in young (P21) WT mice (Fig. 1). In the Fig. 2, the authors show images from 3-m.o. mice, demonstrating reduced spine density only at the distal dendrites of CA1 neurons in J20 mice, displaying the transition from elongated mitochondria (in WT) to fragmented mitochondria (in J20). The question is whether the same decrease in spine density and mitochondrial morphology occurs in L2/3 cortical neurons? It is necessary to show the universality of the phenotype observed in the distal CA1 dendrites.

Technical problems:

1. The authors should also consider that Drp1 may be recruited to the mitochondria without the ability to efficiently perform fission in FAD model (Zhang et al. 2016). Fluorescent microscopy can't resolve this kind of structures, so what for the authors look like smaller round mitochondria could be the effect of a series of incomplete fission events. Therefore, it is important to quantify fission/fusion events after treatment of amyloid-beta. Moreover, along with increased phosphorylation of MFF, the authors could show DRP1 S616 and S637 levels (or Drp1 localization using super-resolution) to give a more complete picture.

2. The previous study (Fang et al., Nature Neuro 2019) shows that pAMPK, pMFF and p616 Drp1 are increased, while mitophagy is downregulated in AD samples and AD iPSC-derived neuron. A better quantification of mitophagy is required to verify the findings (for example by mKeima method). LAMP1 overexpression may marking endosomes, in addition to lysosomes. There is no mitochondrial marker for colocalization in Fig. 4c.

3. Fig. 8b-c and Fig. 9d-e - how do the authors explain the reduction in mitochondria length, without a reduction in mitochondria density, which is the mitochondria length/ dendrite length?

4. Fig.9 - the authors wrote that introduction of WT-ULK2 to ULK2-KO cultures restored amyloid-beta-induced mitochondria fragmentation, but the data in Fig. 9d does not support this claim.

Rebuttal NCOMMS-20-02539

Lee, Kondapalli, Virga et al.

A β 42 oligomers trigger synaptic loss through CAMKK2-AMPK-dependent effectors coordinating mitochondrial fission and mitophagy

Reviewer #2 (Remarks to the Author):

The manuscript by Lee and colleagues builds on previous work from the Polleux group and others to further investigate the pathway leading to mitochondrial alteration and subsequent spine loss. It has previously been illustrated that pathogens, such as AB42o alters AMPK activity and results in mitochondrial fragmentation and abnormal mitochondrial distribution (Thornton et al 2011, Yoon et al 2012, Mairet-Coello et al 2013). In the present study the authors firstly confirm the previous findings and build on this to identify the specific mechanism leading to mitochondrial fission (MFF) and mitophagy (ULK2) which results in spine loss.

While this is advance is modest, by uncovering the full molecular pathway as it relates to the mitochondria, it does uncover potential translational findings; the ability to inhibit the pathways (MFF and ULK2), thereby inhibiting spine loss without altering normal neuronal development or synaptic maintenance. Therefore, opening in possibility as a potential early stage therapeutic target. However, the paper does not show how the mitochondria alterations result in spine loss, merely that they are required for the spine loss to occur. Furthermore, the current manuscript has a limited scope of mechanistic understanding compared with authors previous publication.

We thank this reviewer for their synthesis of our current and previously published work. However, we would like to point out that none of the previously cited (Thornton et al 2011, Yoon et al 2012, Mairet-Coello et al 2013) examined the relationship between Abeta-dependent AMPK over-activation and mitochondria structure/function. This is the novelty of the present work only--previous work from our lab (Mairet-Coello et al. 2013) and others (Thornton et al 2011, Yoon et al 2012) focused on CAMKK2-AMPK signaling and some potential downstream effectors (mostly Tau-S262 phosphorylation), but **neither we nor others had ever causally linked A β 42 oligomer-dependent dendritic mitochondrial remodeling to synaptic maintenance**. Therefore, we think our results have multiple layers of novelty as detailed below.

1. The authors uncover an interesting finding in that the hippocampal CA1 neurons present with a compartmentalized mitochondrial distribution in comparison to the cortical neurons. It would be beneficial to the manuscript if the authors had investigated this difference e.g. do the hippocampal mitochondria respond the same way to pathogens as the cortical neurons, or do they show different responses based on the compartment.

This is potentially illustrated, but not investigated/expanded, in figure 2. In which Abeta treatment does not alter the mitochondria length or density in the apical oblique (CA2/CA3) or basal compartments. This is surprising as previous research has shown reduced LTP upon Abeta incubation which requires schaffer collateral pathway (CA3 – CA1). The authors should explain how the findings of the manuscript fit into the literature regarding this point, and/or complete experiments at older ages.

We appreciate this comment, and we would like to point out that we do provide *in vivo* evidence showing that mitochondrial morphology and spine density is affected at 3 months in the J20/APP^{SWE,IND} mouse model specifically in the apical tuft (corresponding to inputs from entorhinal cortex) but not apical oblique (which receives exclusive inputs from CA3) or basal dendrites (mixed inputs from CA3 and CA2). On top of revealing (for the first time to our knowledge) that mitochondrial morphology is

strikingly compartmentalized *in vivo*, showing elongated/fused morphology only in apical tufts in wild-type CA1 PNs, this observation is foundational to the rest of the article where we reveal (for the first time also) the molecular pathways mediating the effects of A β 42 oligomers on mitochondrial morphology (increased fission through AMPK-dependent MFF over-activation) and mitophagy (through AMPK-dependent ULK2 over-activation).

We would also like to add that, unlike what is suggested by this reviewer, we do not attempt to explain every synaptic phenotype induced by A β 42 oligomers. Rather, we focus on the relationship between CAMKK2-AMPK-MFF-ULK2 dependent mitochondrial remodeling and synaptic maintenance. It is completely plausible that A β 42o exerts its influence on synaptic plasticity (LTP and LTD rather than synaptic maintenance as shown in the current study) in apical obliques or basal dendrites of CA1 PNs through a pathway independent of AMPK as exemplified by many studies. However, recent evidence (cited in Discussion) from the Klann lab {Ma, 2014 #48} and from the Vingtdoux's lab {Domise, 2016 #746; Domise, 2019 #732; Vingtdoux, 2011 #86}.

2. The authors state that the PNs of CA1 show compartmentalization while cortex does not, while this has been shown for juvenile animals (p21 – figure 1) it was not shown for adult animals (p90 – figure 2). Therefore, the authors should clarify in the text the age ranges etc or add adult cortical data.

We clarified the ages examined in the legend and the text. We also referenced a recent pre-print which quantified mitochondrial morphology throughout dendrites of layer 2/3 mouse PNs (Turner et al. (2020) <https://www.biorxiv.org/content/10.1101/2020.10.14.338681v3.full.pdf>) using serial EM and demonstrates that mitochondria are fused throughout entire dendritic arbor of these neurons *in vivo* in adult (P40) mouse.

3. While the authors show spine density reductions it is becoming more apparent that spine type (morphology) is important. Can the authors examine the morphology of the spines, particularly in the hippocampal data, spine changes may have been missed in the apical oblique region by only examining density.

We re-analyzed spine morphology in cortical PNs following treatment with A β 42o and have added a supplemental figure (Fig. S09) detailing those findings. In short, there does appear to be significant differences in the spine morphologies between control neurons and A β 42o-treated neurons, with A β 42o-treated neurons demonstrating a significant reduction in spine width and spine length, which is rescued when knocking down MFF which also prevents spine loss. When qualitatively categorizing the spines, there is a trending but insignificant decrease in mushroom-like spines and increase in thin spines in A β 42o-treated neurons, which has been described previously by many groups.

4. Was spine loss associated with mitochondria changes? – e.g. Did a larger alteration in mitochondria result in greater spine loss (time lapse imaging required)? Or are the mitochondria changes uniform across the whole of the cortical neuron.

We performed correlative analyses between spine density and mitochondrial density within dendritic segments, and added this to a supplemental figure (Fig. S09). For control cortical PNs and A β 42o-treated cortical PNs, we found a significant ($r=0.5998$; $p<0.0001$) positive correlation between spine density and mitochondrial density. This indicates that a reduction in mitochondrial density in a given dendritic segment is directly correlated with a reduction in spine density, while an increase in mitochondrial density is directly correlated with an increase in spine density, regardless of treatment condition, indicating a potential generic neuronal property.

5. In main text line 371, authors discussed “disrupted cytoplasmic Ca $^{2+}$ homeostasis that drives PNs hyperexcitability.....”. It would be better to include and discuss Ca $^{2+}$ -permeable AMPA receptor

hypothesis in Aβ-mediated synapse dysfunction. There is a literature described rapid aberrant calcium permeable AMPA receptor hypothesis (Whitcomb et al, 2015?). This would be one of others which could contribute mitochondria-mediated synapse dysfunction.

We thank this reviewer for their comments and added this and other references as well as discussion points related to the importance of changes in AMPA receptor content following Aβ42 oligomer treatment.

Other points:

In FIG2.D Mitochondria look longer and brighter in this figure

FIG3 A'-C' I can see the reduction of mitochondria in terms of number of elements but they seem brighter, explain this why?

In FIGURE 4 B and C are inverted

In FIGURE 7 all the figure legends the letters indicating the relative images are a mix of capital and small size and not always proper matched.

We have addressed all of the above minor points in the figures, legends, and manuscript.

Reviewer #3 (Remarks to the Author):

In the current manuscript, Lee et al. demonstrated that MFF and ULK2 mediate amyloid-beta-induced structural changes, reflected by reduction in dendritic mitochondrial length/density and synapse loss. These molecular pathways were linked to AMPK-CaMKK2, previously described by the authors to mediate amyloid-beta-induced synapse loss.

The mechanism of synaptic loss is a long-standing question in AD biology. Accumulated evidence suggests that ER-Ca plays a crucial role in this process and many molecular players have been already discovered. The authors stated "For the first time, we demonstrate that Aβ42 mediates loss of excitatory synapses through MFF-ULK2-dependent mitochondrial fission and mitophagy".

Unfortunately, they did not provide the mechanisms linking MFF and ULK2 to synaptic loss.

Mitochondrial fragmentation / reduction in mitochondrial mass, well known AD phenotypes, do not explain why synapses become unstable and disappear. Clearly, the structural mitochondrial changes are associated with functional mitochondrial deficits - such as energy / calcium / redox imbalance - that remain to be identified as a potential cause of synapse loss.

We thank this reviewer for their synthesis of our current work, and for highlighting the importance of uncovering the mechanisms of synaptic loss in AD biology. We agree with the reviewer that, given our findings that MFF-ULK2-dependent reduction in mitochondrial biomass is necessary for spine loss, something never shown before which links two independently observed phenotypes, determining the functional consequences on remaining, fragmented mitochondria, such as their calcium buffering capacity, ATP production, redox levels, etc. is an important next step in this project. However, these experiments are outside the scope of this current publication. We already have 9 main figures and 10 supplementary figures describing a previously unknown causal relationship between Ab42-dependent AMPK-MFF/ULK dependent mitochondrial structural remodeling and synaptic loss which is completely new and we think highly significant. Our discussion section reflects the importance of the current and future experiments for furthering our understanding of the mechanisms underlying synaptic loss in early stages of AD.

MAJOR:

1. The authors previously showed (Lewis et al, 2018) that MFF knockdown (shMFF) causes a reduction

in mitochondrial length exclusively in axons, but not in dendrites in WT neurons. In the same work, shMFF was shown to increase Ca buffering capacity of axonal mitochondria, inhibiting cytosolic Ca transients and synaptic vesicle release. In the submitted manuscript, shMFF occluded amyloid-induced reduction in mitochondrial length and density in dendrites, without having effects on its own in the dendritic compartment. Thus, the main question is how amyloid-beta converts dendritic mitochondria from being MFF-independent to MFF-dependent?

To clarify, our previous work (Lewis et al. 2018) did not show that “MFF knockdown...causes a reduction in mitochondrial length...in axons, but not in dendrites...” Rather, our previous works showed that MFF knockdown increases mitochondrial length in axons, but not in dendrites, as cortical L2/3 dendritic mitochondria are already significantly elongated. This indicates that, in wildtype axons, activated/phosphorylated MFF functions to tip the balance of fission/fusion toward fission, resulting in small, punctate mitochondria only. In dendrites of wild-type layer 2/3 PNs, however, despite MFF being present, it doesn't appear to be activated/phosphorylated to the same extent as axons--a mechanism that is still currently unknown and being investigated--resulting in elongated mitochondria as fusion outweighs fission in this compartment. What “converts dendritic mitochondria from being MFF-independent to MFF-dependent” the evidence described in this manuscript showing at A β 42 through overactivation of CAMKK2-AMPK leads to MFF phosphorylation in the dendritic compartment, tipping the fusion/fission balance toward fission and fragmenting the mitochondria.

There are several possibilities. First, amyloid-beta may affect putative signaling pathways connecting MFF to mitochondrial fragmentation. The second possibility is that alteration of presynaptic function by amyloid-beta precedes synapse loss and these presynaptic alterations constitute the major cause of synapse loss. Therefore, the effects of amyloid-beta on presynaptic functions (Ca and release of synaptic vesicles) and the effects of shMFF on these amyloid-beta-induced presynaptic changes should be tested directly. Previous study shows deficits in density, trafficking and functioning of axonal mitochondria by amyloid-beta (Du et al, PNAS 2010) which may impact synapse loss in vivo.

We agree with the reviewer that presynaptic dysfunction in AD likely contributes to cellular and circuit dysfunction in the disease pathology. **However, we would like to note that all of our manipulations are done strictly postsynaptically, in a cell-autonomous way. Because of this, the inputs to CA1 *in vivo* and the vast majority of inputs to cortical PNs *in vitro* are from genetically unmodified neurons** i.e. wild-type neurons that do not express any plasmids manipulating AMPK, MFF or ULK2. In the case of *in vitro* neurons treated with A β 42, the vast majority of axonal inputs onto an *ex utero* electroporated neuron which has MFF knocked down are originating from unlabeled neurons in which A β 42 is instigating pathology--mitochondrial density reduction, spine reduction, potential calcium balance dysfunction, etc. Despite these potential effects of A β 42 on presynaptic boutons/axonal inputs, knocking down MFF cell-autonomously (i.e. postsynaptically only at such low density) and **therefore preventing mitochondrial biomass loss in dendrites is sufficient to prevent postsynaptic density reduction in electroporated neurons**. The only interpretation of all our analysis is that preventing MFF mediated mitochondrial fission is sufficient to prevent spine loss.

It would be important to test several time points, including those preceding synapse loss. The effect of amyloid-beta on cytosolic and axonal mitochondrial Ca, axonal mitochondrial length/density, as well as on synaptic vesicle release should be quantified. Based on the literature, amyloid-beta may cause a wide range of presynaptic effects, from augmentation to inhibition, depending on its biophysical properties (that's why it would be preferable to make knockdown in J20 animals). The authors should test whether shMFF occludes presynaptic effects of amyloid-beta under the experimental conditions used in the study. Reduction in the number of spines could be secondary to inactivity of presynaptic terminals or a compensatory mechanism to hyperactivity (presynaptic or hyperexcitability).

In the same line, AMPK has been shown to regulate presynaptic function in other brain regions (Yang et al., Cell 2011).

We have now performed *in vivo* experiments (which are more challenging of course) knocking down MFF in WT or J20 mice (**new Figure 6**) demonstrating that **preventing MFF-mediated fission in CA1 PN in vivo has no effect on dendritic spine density in WT mice but completely blocks spine loss in age-matched J20 mice.**

2. Does the effect of shMFF on mitochondrial length depend on ER-Ca? This is a critical question to answer in order to connect the present study to numerous previous findings in the field (the authors simply ignored all of them despite working on Ca-dependent CaMKK2-AMPK pathway). Does amyloid-beta cause ER-Ca overload that leads to overload of mito-Ca? Does shMFF rescue mito-Ca overload? Recent work suggests that amyloid-beta modulates ER-mito Ca crosstalk in hippocampal neurons and this effect is age-dependent (Calvo-Rodriguez et al., 2019). Misregulation of ER-mito Ca crosstalk by amyloid-beta may be reason of synapse loss. What is the role of PDZD8, discovered by the authors, in this mechanism?

We appreciate the reviewer's familiarity with our work in discovering the ER-mitochondria tethering protein, PDZD8, and its necessity in facilitating ER-mitochondrial calcium buffering. We agree that understanding the impact of A β 42o on ER-mitochondria contacts and subsequently on mitochondrial calcium buffering could be an important step to further understanding the mechanisms of synaptic loss in AD biology, and have included the citation suggested by the reviewer in our discussion the implications our study has on further exploration of this question.

It is likely that, because there is a significant reduction in dendritic mitochondrial biomass in CA1 PNS apical tufts in J20 mice and in neurons exposed to A β 42o, there is a subsequent decreased capacity for mitochondria to store Ca²⁺ and therefore increase in cytosolic calcium, further increasing Ca²⁺-dependent CAMKK2-AMPK overactivation as part of a vicious circle, worsening the phenotype over time. However, we strongly think this is beyond the scope of the present study.

3. The authors suggest that mitochondrial fragmentation causes spine loss, while the opposite may be true. For this, they have to demonstrate the time cause of fragmentation vs time-cause of synapse loss *in vitro* and *in vivo*.

The fact that we can completely block spine loss induced by A β 42o *in vitro* and *in vivo* cell autonomously by blocking MFF-dependent fission demonstrates without any other interpretation that the causal relationship links A β 42o-triggered mitochondrial fission to synaptic loss and not the other way around.

4. An alternative possibility is that MFF/ULK2/AMPK affect amyloid-beta levels in J20 mice and this is the main reason for spine loss *in vivo*. The authors should check how knockdown of these proteins affects amyloid-beta levels and amyloid-beta 42/40 ratio and whether it rescues synapse loss *in vivo*. Artificial addition of a mixture of monomeric and oligomeric amyloid-beta 42 preparation that causes synapse loss in 24 hr in cultures is very different from slow process of synapse loss *in vivo* that starts at distal dendrites in 3-m.o. J20 mice, highly likely are evident at many other dendrites at later stages.

We appreciate this suggestion from the reviewer and have since added additional experiments in which A β plaque formation has been measured in 5-month old WT, J20, and J20 in which AMPK α 1 has been constitutively knocked out in a heterozygous and homozygous way, included in Supplemental Figure 10. Interestingly, we do find that removing AMPK α 1 completely (but not in heterozygous mice) leads to a significant reduction in the total hippocampal A β plaque coverage as well as A β plaque size in J20 mice, indicating that AMPK may potentially play an additional role in APP processing. These results

suggest that in vivo, AMPK over-activation observed in patients and mouse models could in part participate to AD progression due to its role in increasing APP processing and A β 42 accumulation. However, our data also demonstrate that upon acute application of A β 42o, preventing over-activation of CAMKK2, AMPK, MFF, ULK2 or TauS262 phosphorylation cell-autonomously protects cortical PNs from synaptic loss, and therefore that a significant part of the effect of over-activating this signaling pathway occurs intracellularly and cell-autonomously in dendrites/neurons.

5. The article begins from showing that in vivo mitochondria morphology in cortical L 2/3 and in CA1 apical tuft dendrites are mostly elongated in young (P21) WT mice (Fig. 1). In the Fig. 2, the authors show images from 3-m.o. mice, demonstrating reduced spine density only at the distal dendrites of CA1 neurons in J20 mice, displaying the transition from elongated mitochondria (in WT) to fragmented mitochondria (in J20). The question is whether the same decrease in spine density and mitochondrial morphology occurs in L2/3 cortical neurons? It is necessary to show the universality of the phenotype observed in the distal CA1 dendrites.

AD pathology originates, both in humans and in our mouse model, in the entorhinal cortex and hippocampus well before progressing to the neocortex. Our unique findings demonstrating preferential loss of mitochondrial biomass in apical tuft of CA1 PNs at 3 months in J20 mice where the first spine loss is observed links the unique early events described in mouse models and AD patients. Furthermore, we now show in the **new Figure 6** that in 3 months old mice, mitochondria found in apical dendrites of WT are elongated but fragmented in J20 mice, we also demonstrate that this is prevented by preventing MFF-dependent mitochondrial fission in J20 mice.

Technical problems:

1. The authors should also consider that Drp1 may be recruited to the mitochondria without the ability to efficiently perform fission in FAD model (Zhang et al. 2016). Fluorescent microscopy can't resolve this kind of structures, so what for the authors look like smaller round mitochondria could be the effect of a series of incomplete fission events. Therefore, it is important to quantify fission/fusion events after treatment of amyloid-beta. Moreover, along with increased phosphorylation of MFF, the authors could show DRP1 S616 and S637 levels (or Drp1 localization using super-resolution) to give a more complete picture.

We appreciate this suggestion, and while it would be interesting to examine the role of Drp1 recruitment in MFF hyperactivation, we believe this is beyond the scope of the present study.

2. The previous study (Fang et al., Nature Neuro 2019) shows that pAMPK, pMFF and p616 Drp1 are increased, while mitophagy is downregulated in AD samples and AD iPSC-derived neuron. A better quantification of mitophagy is required to verify the findings (for example by mKeima method). LAMP1 overexpression may marking endosomes, in addition to lysosomes. There is no mitochondrial marker for colocalization in Fig. 4c.

We greatly appreciate the reviewer for bringing this paper to our attention which we now discuss extensively. We performed experiments using both mito-mKeima and mito-QC as alternative methods for measuring mitophagy and found they are utterly less sensitive than previously reported and less reliable than our coalescence of LAMP1+ lysosomes and LC3+ autophagosomes at sites where mitochondrial biomass is lost. We decided not to include these data.

3. Fig. 8b-c and Fig. 9d-e - how do the authors explain the reduction in mitochondria length, without a reduction in mitochondria density, which is the mitochondria length/ dendrite length?

Because MFF is still hyperactive in these cells, fission outweighs fusion and individual mitochondria become smaller. However, **because we are preventing ULK2-dependent mitophagy, there is no reduction in overall mitochondrial biomass.** This result showing that reduction in mitochondrial length but not density/biomass is not sufficient to induce spine density, strengthens our argument that fission and subsequent mitophagy are a coupled mechanism, ultimately resulting in a reduction of mitochondrial biomass which is responsible for spine loss (not simply increased fission).

4. Fig.9 - the authors wrote that introduction of WT-ULK2 to ULK2-KO cultures restored amyloid-beta-induced mitochondria fragmentation, but the data in Fig. 9d does not support this claim.

We have edited this mistake in the manuscript. Introduction of WT-ULK2 in ULK2-deficient cortical PNs does not prevent fragmentation and subsequent reduction in mitochondrial density. Rather, introduction of ULK2-KI (kinase inactive) or ULK2-4SA (a form of ULK2 that cannot be phosphorylated by AMPK) prevents mitochondrial density reduction and prevents spine loss induced by A β 42o.

Reviewers' Comments:

Reviewer #3:

Remarks to the Author:

The revised version of the manuscript includes a new Figure 6 that shows the role of MFF in mitochondrial fragmentation and spine loss in J20 model. However, many serious problems remain unresolved.

First, the novelty of this work is limited. All the molecular players – Aβeta, tau, MFF, ULK and AMPK were previously described in AD-related synaptic and behavioral deficits.

Second, the results of this work are not confirmed by other AD transgenic models and human AD brain samples. In this paper, the authors show that mitochondria regulate synapse loss in the distal part of basal CA1 PN dendrites (in SLM area). The authors explain it by a different fission-fusion regulation in SLM dendrite (showing spaghetti-like mitochondria) in comparison to all other dendritic regions, comprising the majority of the dendritic tree (showing round mitochondria). The authors found that it is only distal apical dendrites that show reduced spine density at the early stage in J20 mice. However, previous studies suggest that it is not the case in different transgenic mouse models of AD. For example, in less aggressive Tg2576 model, synaptic loss is initiated in proximal, but not in distal CA1 PN dendrites (Perez-Cruz et al J Neurosci 2011). In the same paper, in APP/Lo models, synaptic loss is stronger in proximal than in distal CA1PN dendrites. The citation of this paper is misleading and the discrepancy in results are not discussed. Thus, the results of J20 model are not reproducible in other models and may be due to very strong APP overexpression. The results related to synthetic Ab42 oligomer application are not valid by its own – today it is widely accepted that the relevance of these experiments to AD progression is questionable and most of unreproducible / controversial results in the literature are due to different concentrations / conformations of Ab42 used by different studies and their relevance to AD biology remains questionable.

Third, the conclusions of the current manuscript contradict Fang et al (Nature Neuroscience 2019, >500 citations) who used more extensive characterization of mitochondrial function and mitophagy in 3 different AD models. The authors provide an explanation for this discrepancy which I cannot accept. Fang et al show similar results in iPSC-derived neurons and in brain tissue, suggesting that glial cells are NOT the source of the controversy. In my opinion, the quantification of mitophagy by 'mitophagy index', defined by the authors as "the intensity of mito-mTagBFP2 fluorescence using line scan intensity measurements in regions of interest (ROI) with or without LC3+/LAMP1+ puncta" is not convincing, and significant bleaching at t=0-200 min in Fig. 3E is higher than the difference between control and Ab42 at t=400-800 min (all the difference is <5%). The authors wrote that they did not get robust results with 2 additional methods used by others and decided not to include these results. The lack of effect by other methods may imply that the true difference is very small or the 'mitophagy index' does not provide a true measurement of mitophagy. The authors did not show any positive control which is typically used in mitophagy studies (for example mitophagy induction by ETC blockers) and they did not show any quantification of mitochondrial function / dysfunction as has been extensively done by Fang et al.

In summary, the general conclusions on the mechanism of AD-related spine loss made by this paper are misleading and at best are relevant for the distal part of CA1 PN dendrites in J20 model.

Reviewer #4:

Remarks to the Author:

This paper examines the role of AMPK activation in AD models (J20 mice and Ab42 treated neurons), and finds that AMPK phosphorylation of Mff contributes to mitochondrial fission, while AMPK phosphorylation of ULK2 at 4 potential new phosphorylation sites leads to mitophagy, which may contribute to decreases in spine density in these models. Overall, this is a very interesting paper which links AMPK activation in AD with the role of AMPK phosphorylation of Mff and ULK2. The authors do a nice job of examining how mutations in Mff and ULK2 which cannot be phosphorylated by AMPK can rescue defects in mitochondrial and spine density. Some parts are a

little overstated and the conclusions drawn from the mitophagy experiments could be better clarified, as detailed below.

The authors did a good job responding to Reviewer #2's comments:

- They have clarified the animal ages, analyzed spine morphology and association of spine loss with mitochondrial changes (Fig S9), mentioned the role of calcium in the Discussion, and also addressed minor comments.

- Their response to Reviewer #2's first comment (hippocampal vs cortical neurons) could be addressed a little more in the discussion (it was unclear if they did, as their response appears to be incomplete in their last sentence response)

Additional Comments:

1. In Fig 3A' (reduced mito fluorescence in area close to LC3 puncta) – it is unclear what this means and this image should be potentially removed. The examples at 0 hour (or at 14 hour) for Ab42 do not look as if there is a blue mitochondria inside the LC3/Lamp1 vesicle. Similarly, the kymographs show loss of signal for mitochondria, but one would expect a mitochondrial signal that is visibly separate at 0 hour where the yellow arrows are (rather than merged with the background mitochondria signal). The fragmentation of mitochondria treated with Ab42 at 14 hours also look more like bulging and fission of mitochondria over time in unhealthy cells (due to ab42 treatment) rather than mitophagy events.

2. The calculation of the mitophagy index (Fig 3F) is a little misleading. It assumes that the presence of an autophagosome indicates that it previously engulfed a mitochondria in the neighbouring area (whereas the actual presence of a clear mitochondrial puncta inside an autophagosome would be the clearest indicator), and further assumes that changes in mitochondrial density are the direct result of a mitophagy event in that localized area. This does not take into account: 1) mitochondrial fission/fusion events which might change the mitochondrial density over time, 2) the ability of autophagosomes to move within dendrites over 14 hours, and 3) the ability of mitochondria to move within dendrites during that time which can also change the mitochondrial density. The authors may want to mention this in the Discussion or in the Results section (to better clarify) or change the name of their mitophagy index.

3. The authors do not present data on ULK2's direct role on mitophagy in these neurons using LC3/Lamp1/mito (they only use measurements of mitochondrial biomass, which can occur independently of mitophagy). There are also no studies on upstream markers of autophagy (prior to LC3/Lamp1), which one might expect to be altered if ULK2 is disrupted. Of note, defects in ULK2 would affect all types of autophagy (i.e. not specific to mitophagy), which might explain the general increase in autophagosome formation they see in their AD models (which do not always contain clear mitochondria). If the authors do not plan to examine this further in this paper, they may want to soften their claims on mitophagy (and instead mention general autophagy defects) in the Results and Discussion.

4. The authors may want to mention in the Discussion that their analysis does not examine mitochondrial function which could be measured by TMRE/ mitoSOX etc, but only mitochondrial density. It may be interesting to know whether these are also altered in these AD models.

5. There is no evidence for how htau participates in a Mff/ULK2 dependent pathway, since they could be independent: Line 420 should be removed: "This result strongly suggests that AMPK-dependent Tau phosphorylation on S262 participates in the MFF ULK2-dependent mitochondrial remodeling and synaptic loss triggered by A β 42". The authors did a better job discussing this in the Discussion (line 484).

6. For figures (ex. Fig 2) with only 3 bar graphs, the significance comparisons could be drawn out, rather than color coded.

Reviewer #3 (Remarks to the Author):

The revised version of the manuscript includes a new Figure 6 that shows the role of MFF in mitochondrial fragmentation and spine loss in J20 model. However, many serious problems remain unresolved.

First, the novelty of this work is limited. All the molecular players – Aβeta, tau, MFF, ULK and AMPK were previously described in AD-related synaptic and behavioral deficits.

As we mentioned in our previous response, we disagree with this judgement about lack of novelty. Yes of course, Aβ, Tau, AMPK have already been implicated in AD pathophysiology but not MFF and ULK2. Our study is the first to provide causal evidence for the functional requirement of MFF and ULK2 downstream of AMPK overactivation induced by Aβ42 oligomers and even more importantly for the ability of Aβ42 oligomers to trigger synaptic loss both in vitro and in vivo in mouse cortical neurons.

Second, the results of this work are not confirmed by other AD transgenic models and human AD brain samples. In this paper, the authors show that mitochondria regulate synapse loss in the distal part of basal CA1 PN dendrites (in SLM area). The authors explain it by a different fission-fusion regulation in SLM dendrite (showing spaghetti-like mitochondria) in comparison to all other dendritic regions, comprising the majority of the dendritic tree (showing round mitochondria). The authors found that it is only distal apical dendrites that show reduced spine density at the early stage in J20 mice. However, previous studies suggest that it is not the case in different transgenic mouse models of AD. For example, in less aggressive Tg2576 model, synaptic loss is initiated in proximal, but not in distal CA1 PN dendrites (Perez-Cruz et al J Neurosci 2011). In the same paper, in APP/Lo models, synaptic loss is stronger in proximal than in distal CA1PN dendrites. The citation of this paper is misleading and the discrepancy in results are not discussed. Thus, the results of J20 model are not reproducible in other models and may be due to very strong APP overexpression. The results related to synthetic Ab42 oligomer application are not valid by its own – today it is widely accepted that the relevance of these experiments to AD progression is questionable and most of unreproducible / controversial results in the literature are due to different concentrations / conformations of Ab42 used by different studies and their relevance to AD biology remains questionable.

First, regarding the specificity of the spine/synaptic loss in the apical tufts (innervated specifically by entorhinal cortex axons) in J20 versus other mouse models, we now added a Supplementary Table listing all the different mouse models that have been examined for spine/synaptic loss at different time points and in different parts of CA1 PNs. First of all, the paper mentioned by this reviewer is of very poor quality. The spine density is measured on Golgi stained sections (which prevents confocal imaging) and the images shown in the paper are of such poor quality that it is not even comparable with the in vivo images we provide or the images in the other papers we highlighted in the Supplementary Table (Siskova et al. Neuron 2014 and Neuman et al. Brain Struct. Funct. 2015) which both report specific spine loss **in two other mouse models** (respectively APP/PS1 and 5XFAD lines- both more aggressive than the J20 lines) **specifically in the apical tufts of CA1 PNs**. We now added a discussion point in the main text to better put our findings in perspective with the rest of the literature (we also added a Supplementary Table listing all the papers we could find on this topic).

Second, we would like to point out that our conclusions do not only rely on the use of the J20 mouse model in vivo or the use of acute applications of recombinant Aβ1-42 oligomers in vitro. **We went to great length and effort to develop a new human ES cell line where we**

engineered the Swedish mutation in APP to produce human ‘pyramidal-like’ neurons expressing endogenous levels of A β 1-42 oligomers and show in Figure S3 that human neurons derived from these APP^{SWE/SWE} knockin hES cells display strikingly similar mitochondrial fragmentation as observed in the J20 mouse model and also show significant AMPK over-activation measured biochemically. This reviewer might have missed this important addition that we think adds significantly to the strength of our conclusion since they are obtained (1) in human neurons and (2) do not rely on over-expression or synthetic addition of non-physiological levels of A β 1-42 oligomers.

Third, the conclusions of the current manuscript contradict Fang et al (Nature Neuroscience 2019, >500 citations) who used more extensive characterization of mitochondrial function and mitophagy in 3 different AD models. The authors provide an explanation for this discrepancy which I cannot accept. Fang et al show similar results in iPSC-derived neurons and in brain tissue, suggesting that glial cells are NOT the source of the controversy.

We do not think there is substantial controversy or contradiction between our results and the results of Fang et al. The two papers are fundamentally different: all our manipulations are completely cell-autonomous to neurons because all our rescue experiments following acute (24h or less) A β 42 oligomer treatment in vitro or in vivo in J20 mouse model at early time points (3 months in this mouse model corresponds to before any plaque accumulation i.e. at a time point where only are performed using sparse deletion/knockdown of AMPK α 1/ α 2 or MFF or ULK2 only in cortical layer 2/3 PNs or CA1 PNs. All the manipulations done in Fang et al. 2019 rely either on observations in AD patient brains at advanced stages of the disease progression or mouse models past 6 months in more aggressive models than our J20.

Beyond the cell non-autonomous effects of mitophagy promoting treatments such as NAD⁺ supplementation, use of poorly specific UA or AC compounds, one other potential explanation for the apparent discrepancy could be timing: our results demonstrate that at early stages of the disease progression, A β 1-42 oligomers trigger increased ULK2-dependent mitophagy and that preventing ULK2 phosphorylation completely protects cortical PNs from A β 42o-induced spine loss. However, at late stages of the disease, it is possible as shown convincingly by Fang et al. 2019 that neurons become dysfunctional in their ability to promote mitophagy to clear damaged mitochondria.

On the other hand, we cannot exclude that this protecting effect of ULK2 deletion or preventing its phosphorylation by AMPK is mediated not only by blocking ULK2-dependent mitophagy but more generally ULK2-dependent autophagy of other cargoes.

We further modified our Discussion to mention these possible alternative interpretations and also improve our discussion of the Fang et al. 2019 paper.

In my opinion, the quantification of mitophagy by ‘mitophagy index’, defined by the authors as “the intensity of mito-mTagBFP2 fluorescence using line scan intensity measurements in regions of interest (ROI) with or without LC3+/LAMP1+ puncta” is not convincing, and significant bleaching at t=0-200 min in Fig. 3E is higher than the difference between control and Ab42 at t=400-800 min (all the difference is <5%). The authors wrote that they did not get robust results with 2 additional methods used by others and decided not to include these results. The lack of effect by other methods may imply that the true difference is very small or the ‘mitophagy index’ does not provide a true measurement of mitophagy. The authors did not show any positive control which is typically used in mitophagy studies (for example mitophagy induction by ETC blockers) and

they did not show any quantification of mitochondrial function / dysfunction as has been extensively done by Fang et al.

The authors of the Fang et al. 2019 paper used even 'cruder' mitophagy index as their main cellular readouts such as in Fig. 1d simple co-localization of mitochondria and LAMP2 a lysosomal marker and not co-localization of autophagosomes as we did (LC3) and lysosomes (LAMP1). Therefore, we strongly think that the criticism offered by the reviewer is unjustified.

Regarding our mitophagy index, it is very simple and internally controlled: we basically quantify the loss of mitochondrial fluorescence (mito-mTagBFP2) at versus outside areas of dendrites where we see coalescence of LC3 and LAMP1 (marking fused auto-lysosomes). Our quantification in Fig. 3F details that there is basically no loss of mitochondrial biomass during 14h post treatment with A β 42 oligomers at areas of dendrites negative for LAMP1-LC3. This is very strong and simple evidence that A β 42 oligomers promotes not only accumulation of autophagosomes and lysosomes but also that at points along the dendrites where lysosomes and autophagosomes coalesce, mitochondria biomass is significantly decreased. We cannot think of a better index to define precisely the spatio-temporal dynamics of the mitophagy.

What we mentioned in our previous rebuttal regarding the use of two other 'mitophagy' reporters (mito-QC and mito-Keima) is that we could not validate that they are sensitive enough to detect mitophagy in neurons. Most studies published using these reporters use very drastic treatments to induce massive amount of mitophagy such as uncouplers (FCCP). But the mitophagy levels induced by A β 1-42 oligomers in cortical or hippocampal neurons are much less pronounced and require a sensitive way to detect mitophagy provided by our method which is why we developed this much more direct way to assess mitophagy that simply rely on the coalescence of autophagosomes, lysosomes and simultaneously the degradation of mitochondria at that location.

Reviewer #4 (Remarks to the Author):

This paper examines the role of AMPK activation in AD models (J20 mice and Ab42 treated neurons), and finds that AMPK phosphorylation of Mff contributes to mitochondrial fission, while AMPK phosphorylation of ULK2 at 4 potential new phosphorylation sites leads to mitophagy, which may contribute to decreases in spine density in these models. Overall, this is a very interesting paper which links AMPK activation in AD with the role of AMPK phosphorylation of Mff and ULK2. The authors do a nice job of examining how mutations in Mff and ULK2 which cannot be phosphorylated by AMPK can rescue defects in mitochondrial and spine density. Some parts are a little overstated and the conclusions drawn from the mitophagy experiments could be better clarified, as detailed below.

The authors did a good job responding to Reviewer #2's comments:
- They have clarified the animal ages, analyzed spine morphology and association of spine loss with mitochondrial changes (Fig S9), mentioned the role of calcium in the Discussion, and also addressed minor comments.

- Their response to Reviewer #2's first comment (hippocampal vs cortical neurons) could be addressed a little more in the discussion (it was unclear if they did, as their response appears to be incomplete in their last sentence response)

We thank this reviewer for her/his comments on the main findings of our paper. We corrected the manuscript and added a sentence in the Discussion justifying why we used hippocampal neurons in vivo and cortical pyramidal neurons in vitro. The main reason is because cortical PNs in vivo and in vitro display a tubular mitochondria morphology throughout the entire dendritic arbor

(comparable to the mitochondria found only in the distal apical tufts of CA1 PNs in vivo). Therefore, making the assessment of the effects of A β 42o treatments on dendritic mitochondrial morphology much easier. **We clarified this point in the revised text.**

Additional Comments:

1. In Fig 3A' (reduced mito fluorescence in area close to LC3 puncta) – it is unclear what this means and this image should be potentially removed. The examples at 0 hour (or at 14 hour) for Ab42 do not look as if there is a blue mitochondria inside the LC3/Lamp1 vesicle. Similarly, the kymographs show loss of signal for mitochondria, but one would expect a mitochondrial signal that is visibly separate at 0 hour where the yellow arrows are (rather than merged with the background mitochondria signal). The fragmentation of mitochondria treated with Ab42 at 14 hours also look more like bulging and fission of mitochondria over time in unhealthy cells (due to ab42 treatment) rather than mitophagy events.

We thank this reviewer for these remarks, they helped us provide more details and improve our illustrations. We actually re-analyzed our time-lapse video files to extract clear examples of coalescence between autolysosomes and mitochondria and now provide a detailed example of a mitophagy event where an autolysosome engulfs mitochondria, rapidly following a fission event (**new panel Fig. 3A'''**). The kymographs are confusing in a way because the vertical dimension spans 14h (the entire duration of our time-lapse) as illustrated by our frame-by-frame analysis, the mito-mTagBF2 fluorescence inside the autolysosomes does not last more than a few frames (15-30min or so). We think the new panel Fig. 3A''' makes our data more convincing.

2. The calculation of the mitophagy index (Fig 3F) is a little misleading. It assumes that the presence of an autophagosome indicates that it previously engulfed a mitochondria in the neighboring area (whereas the actual presence of a clear mitochondrial puncta inside an autophagosome would be the clearest indicator), and further assumes that changes in mitochondrial density are the direct result of a mitophagy event in that localized area. This does not take into account: 1) mitochondrial fission/fusion events which might change the mitochondrial density over time, 2) the ability of autophagosomes to move within dendrites over 14 hours, and 3) the ability of mitochondria to move within dendrites during that time which can also change the mitochondrial density. The authors may want to mention this in the Discussion or in the Results section (to better clarify) or change the name of their mitophagy index.

We respectfully disagree with this point. First see point above for illustration of the fact that we do indeed observed mito-mTagBF2 fluorescence transiently inside the autolysosomes (coalescence of LAMP1+LC3). Second, We clearly have multiple levels of evidence showing that AMPK-dependent MFF fission occurs in response to A β 42o treatment but that interestingly, when blocking mitophagy mediated by ULK2 (Figure 8) and more precisely AMPK-dependent phosphorylation of ULK2 (Figure 9), MFF-dependent fission induced by A β 42o application still occurs (clear fragmentation occurring) but we do not detect any appreciable decrease in mitochondrial density/biomass (see quantification in Fig. 3C). In other words, our imaging methods and causal molecular interventions clearly allow us to distinguish between fission events and fission+mitophagy events.

3. The authors do not present data on ULK2's direct role on mitophagy in these neurons using LC3/Lamp1/mito (they only use measurements of mitochondrial biomass, which can occur independently of mitophagy). There are also no studies on upstream markers of autophagy (prior to LC3/Lamp1), which one might expect to be altered if ULK2 is disrupted. Of note, defects in ULK2 would affect all types of autophagy (i.e. not specific to mitophagy), which might explain the

general increase in autophagosome formation they see in their AD models (which do not always contain clear mitochondria). If the authors do not plan to examine this further in this paper, they may want to soften their claims on mitophagy (and instead mention general autophagy defects) in the Results and Discussion.

We thank this review for these remarks. We previously performed experiments that we did not think about including before validating the induction of autophagy by A β 42o application and our ability to block this increased autophagy (detected by p62 immunofluorescence over the soma of cortical PNs in vitro) by knocking down ULK2. This new data is shown in Fig. S7 and we think improve our validation of the requirement of ULK2 for A β 42o-induced autophagy but also reinforces the valid point raised by this reviewer that we cannot formally exclude the possibility that AMPK-dependent ULK2 over-activation could mediate its effects on spine loss induced by A β 42o through increased autophagy of other cargoes than mitochondria. **We modified the Discussion to mention this point.**

4. The authors may want to mention in the Discussion that their analysis does not examine mitochondrial function which could be measured by TMRE/ mitoSOX etc, but only mitochondrial density. It may be interesting to know whether these are also altered in these AD models.

Yes we included this discussion point in the revised text. We also cite several papers already reporting that mitochondrial functions including mitochondrial membrane potential and oxidative phosphorylation are affected in various AD models.

5. There is no evidence for how htau participates in a Mff/ULK2 dependent pathway, since they could be independent: Line 420 should be removed: "This result strongly suggests that AMPK-dependent Tau phosphorylation on S262 participates in the MFF ULK2-dependent mitochondrial remodeling and synaptic loss triggered by A β 42o". The authors did a better job discussing this in the Discussion (line 484).

We modified the text to mention this point. We also cited papers recently showing that Tau participates in the execution of various forms of autophagy.

6. For figures (ex. Fig 2) with only 3 bar graphs, the significance comparisons could be drawn out, rather than color coded.

We find that when multiple comparisons are displayed even for 3 groups only, the statistical analysis gets very busy so we prefer to keep the color coding to indicate which groups are compared.

Reviewers' Comments:

Reviewer #4:

Remarks to the Author:

The authors have addressed most of the comments previously made:

- Fig 3A is improved, with a clearer example of fission and mitophagy
- We appreciate the data in Fig S7 on ULK2 and for softening the mention of ULK2's role in mitophagy in the discussion.
- We appreciate the authors editing the discussion points as requested

In the future, it will be helpful for the authors to include the revised manuscript with highlighted changes in the discussion text (or point out which paragraph/line/page in discussion the changes were made in the response letter). Similarly, it will be helpful to number the main figure # at the top of each figure, since it is hard to keep track of each figure without numbering. The authors may also want to flatten both main/supp figures prior to submission.

The authors responses to Rev #3 regarding differences with previous data is sufficiently explained and with their included supp table. However, as both reviewers commented on potential limits of the mitophagy index, the authors should include several sentences in the discussion mentioning potential limits of this index, prior to publication.

Final revisions for Nature Communications manuscript NCOMMS-20-02539B

Reviewer #4 (Remarks to the Author):

The authors have addressed most of the comments previously made:

- Fig 3A is improved, with a clearer example of fission and mitophagy
- We appreciate the data in Fig S7 on ULK2 and for softening the mention of ULK2's role in mitophagy in the discussion.
- We appreciate the authors editing the discussion points as requested

In the future, it will be helpful for the authors to include the revised manuscript with highlighted changes in the discussion text (or point out which paragraph/line/page in discussion the changes were made in the response letter). Similarly, it will be helpful to number the main figure # at the top of each figure, since it is hard to keep track of each figure without numbering. The authors may also want to flatten both main/supp figures prior to submission.

The authors responses to Rev #3 regarding differences with previous data is sufficiently explained and with their included supp table. However, as both reviewers commented on potential limits of the mitophagy index, the authors should include several sentences in the discussion mentioning potential limits of this index, prior to publication.

Response:

We thank this reviewer for his/her support. We added yet another Discussion point to address the potential limitations of our mitophagy index on Page 13:

We developed a direct way to visualize mitophagy using quantifications of mitochondria 'degradation' in time-lapse analysis where genetically-encoded mitochondrial markers disappeared in location where LAMP1+ lysosome and LC3+ autophagosome fused. We computed a 'mitophagy index' which reflects mitochondria biomass loss at location where autophagolysosomes emerge, however, this index might also be influenced by local mitochondria dynamics including fission events followed by mitochondria transport independent of mitophagy events. Despite this potential limitation, our rescue experiments in PNs performed by gene replacement strategies reinforces that MFF-dependent fission coupled with ULK2-dependent autophagy is causally linked with dendritic spine loss induced by A β 42o. Altogether, our results strongly argue that at least during early stages of the disease progression, A β 42o-induced over-activation of CAMKK2-AMPK mediates synaptic loss through drastic structural remodeling and loss of biomass of dendritic mitochondria.